# Extratropical cyclone induced sea surface temperature anomalies in the 2013/14 winter

Helen F. Dacre [1], Simon A. Josey [2], and Alan L. M. Grant [1]

[1]University of Reading
[2]National Oceanography Centre

**Correspondence:** H. F. Dacre (h.f.dacre@reading.ac.uk)

**Abstract.** The 2013/14 winter averaged sea surface temperature (SST) was anomalously cool in the mid-North Atlantic region. This season was also unusually stormy with extratropical cyclones passing over the mid-North Atlantic every 3 days. However, the processes by which cyclones contribute towards seasonal SST anomalies are not fully quantified. In this paper a cyclone identification and tracking method is combined with ECMWF atmosphere and ocean reanalysis fields to calculate cyclone-relative net surface heat flux anomalies and resulting SST changes. Anomalously large negative heat flux is located behind the cyclones cold front resulting in anomalous cooling up to 0.2K/day when the cyclones are at maximum intensity. This extratropical cyclone induced 'cold wake' extends along the cyclones cold front but is small compared to climatological variability in the SST's. To investigate the potential cumulative effect of the passage of multiple cyclone induced SST cooling in the same location we calculate Earth-relative net surface heat flux anomalies and resulting SST changes for the 2013/2014 winter period. Anomalously large winter averaged negative heat flux occurs in a zonally orientated band extending across the North Atlantic between 40-60°N. The 2013/2014 winter SST cooling anomaly associated with air-sea interactions (anomalous heat flux, mixed layer depth and entrainment at the base of the ocean mixed layer) is estimated to be -0.67 K in the mid-North Atlantic ($68\%$ of the total cooling anomaly). The role of cyclones is estimated using a cyclone masking technique which encompasses each cyclone centre and its cold wake. The environmental flow anomaly in 2013/2014 sets the overall tripole pattern of heat flux anomalies over the North Atlantic. However, the presence of cyclones doubles the magnitude of the negative heat flux anomaly in the mid-North Atlantic. Similarly, the environmental flow anomaly determines the location of the SST cooling anomaly but the presence of cyclones enhances the SST cooling anomaly. Thus air-sea interactions play a major part in determining the extreme 2013/2014 winter season SST cooling anomaly. The environmental flow anomaly determines where anomalous heat flux and associated SST changes occur and the presence of cyclones influences the magnitude of those anomalies.

## 1 Introduction

The interaction of the ocean and atmosphere has long been recognised as an important element of oceanic cyclogenesis. In the tropics, sub-saturation of air above the sea surface in the vicinity of tropical cyclones results in strong surface heat flux which cools the upper ocean in the wake of tropical cyclones (Kleinschmidt, 1951; Fisher, 1958). In addition, strong winds

enhance mixing in the upper-ocean, which can result in the transport of cool water to the surface. Similarly, in the mid-latitudes it was observed by Pettersen et al. (1962) that surface heat fluxes are largest in the advancing cold air mass behind an extratropical cyclone's cold front. Although fluxes are typically an order of magnitude smaller in extratropical cyclones than tropical cyclones, they still have the potential to influence the underlying ocean.

Alexander and Scott (1997) analyzed the association of ocean heat fluxes with propagating extratropical cyclones on synoptic timescales. They found increased fluxes directed from the ocean to the atmosphere in the western parts of the cyclones and the eastern parts of the cyclones were associated with decreased fluxes directed from the ocean to the atmosphere. These results have been confirmed by many subsequent studies (Persson et al., 2005; Nelson et al., 2014; Schemm and Sprenger, 2015; Dacre et al., 2019) and suggest a close association between the cyclones and surface turbulent fluxes in the midlatitudes. However, a cyclone compositing study by Rudeva and Gulev (2011) found that although composites of fluxes show locally very strong positive fluxes in the rear part of the cyclone, the total air-sea turbulent fluxes provided by cyclones were not significantly different from the averaged background fluxes in the North Atlantic suggesting at least partial cancellation of the flux anomalies associated with cyclones.

The anomalous surface heat fluxes generated by cyclones can create SST anomalies known as the 'cold wake' effect. Case studies of winter cyclones in the North-West Atlantic have found SST cooling in the rear part of cyclones of between 0.4 and 2 K (Ren et al., 2004; Nelson et al., 2014; Kobashi et al., 2019). This is largely due to enhanced turbulent fluxes behind the cold front, however Kobashi et al. (2019) also attribute part of the cooling to cloud shielding of incoming solar radiation, although this is possibly due to the more southerly latitude of the cyclone in their study. Cooling may also result from an episodic wind effect, known as resonant wind-driven mixing, which occurs when the rotation rate of the winds at a fixed point matches the rotation of the wind driven currents (Crawford and Large, 1996). The magnitude of the cooling has been shown to depend on the cyclone's intensity (Yao et al., 2008; Rudeva and Gulev, 2011) with stronger cyclones creating enhanced surface fluxes and hence increased cooling. In addition, there is some seasonality in the cooling magnitude, with largest cooling occurring in late summer and autumn when the ocean surface mixed layer in the Northern Hemisphere is shallower (Ren et al., 2004; Kawai and Wada, 2011). Finally, the relationship between enhanced turbulent fluxes is regionally dependent. For example, Tanimoto et al. (2003) showed that in the central North Pacific, enhanced turbulent fluxes can generate local SST variations but in regions where ocean dynamics are important, such as the western Pacific, the SST anomalies formed in the early winter determine the mid- and late-winter turbulent heat flux anomalies rather than the passage of cyclones. Similarly, Buckley et al. (2015) find that in the Gulf Stream region, ocean dynamics are important in setting the upper-ocean heat content anomalies on interannual time scales and that air–sea heat fluxes damp anomalies created by the ocean.

The effect of extratropical cyclone induced fluxes on longer timescale variability has been investigated in both the Atlantic and Pacific. Several studies have shown that wintertime fluxes in the Gulf Stream are characterized by episodic high flux events due to cold air outbreaks from North America associated with the passage of extratropical cyclones (Zolina and Gulev, 2003; Shaman et al., 2010; Parfitt et al., 2016, 2017; Ogawa and Spengler, 2019). Similar relationships have been found in the high-latitude South Pacific by Papritz et al. (2015). The influence of these enhanced fluxes is to restore the low-level atmospheric

baroclinicity destroyed by the passage of the cyclones (Papritz and Spengler, 2015; Vannière et al., 2017) but the impact on the underlying SST's has not been quantified.

While the role of individual cyclones on local SST's have been studied, the cumulative effect of cyclone induced SST changes over individual seasons has not received much attention in the literature. During the winter of 2013/2014 a cold anomaly developed in the SST in the mid Atlantic. Grist et al. (2016) found that during this winter enhanced sensible and latent heat fluxes occurred in the North Atlantic, with latent heat fluxes being largest in the east North Atlantic and sensible heat flux anomalies stronger in the west North Atlantic resulting in a reduction in ocean heat content of the subpolar gyre. The extent to which extratropical cyclones were responsible for these enhanced fluxes and associated cooling is not well understood.

In this paper we investigate both the SST cooling associated with individual cyclones and the SST cooling associated with the passage of multiple cyclones over the same location in the 2013/2014 season to determine how significant cyclones were in contributing to the observed cooling.

## 2 Data and Analysis Methods

### 2.1 ERA-Interim data

ERA-Interim is a global atmospheric reanalysis dataset (Dee et al., 2011). The data assimilation system used to produce ERA-Interim is based on Integrated Forecasting System (Cy31r2). The system includes a 4-dimensional variational analysis with a 12 hour analysis window. The spatial resolution of the data set is approximately 80 km (T255 spectral) on 60 vertical levels from the surface up to 0.1 hPa. 6 hourly ERA-Interim data has been used in this study to determine extratropical-cyclone related SST changes. We analyse several re-analysis fields from ERA-Interim which are described in this section.

The net surface thermal radiation ($Q_{LW}$) is the thermal radiation emitted by the atmosphere and clouds reaching the surface minus the amount emitted by the surface. Surface solar radiation ($Q_{SW}$) is the amount of solar radiation reaching the surface (both direct and diffuse) minus the amount reflected by the surface. Surface latent heat flux ($Q_E$) is the exchange of latent heat with the surface through turbulent diffusion and the surface sensible heat flux ($Q_H$) is the exchange of sensible heat with the surface. The magnitudes of $Q_E$ and $Q_H$ depend on the windspeed and moisture and temperature differences between the surface and the lower atmosphere.

The net surface heat flux ($Q_N$) is given by the sum of $Q_{SW}$, $Q_{LW}$, $Q_H$ and $Q_E$. The ECMWF convention for vertical fluxes is positive downwards. We also analyse ERA-Interim SST's which are the temperatures of sea water near the surface. ERA-Interim SST's are taken from different sources depending on the dates of the reanalyses: NCEP 2D-Var SST (Jan 1989 – Jun 2001); NOAA Optimum Interpolation SST v2 (Jul 2001 – Dec 2001); NCEP Real-Time Global SST (Jan 2002 – Jan 2009); Met Office Operational SST (Feb 2009 - 2015).

## 2.2 ECMWF Ocean Reanalysis System (ORAS5)

The ECMWF Ocean Reanalysis System 5 (ORAS5) is a global eddy-permitting ocean-sea ice reanalysis system. It provides an estimate of the historical ocean state from 1979 to present. The ocean model resolution in ORAS5 is 0.25 degree in the horizontal (approximately 25 km in the tropics, and increasing to 9 km in the Arctic) and 75 levels in the vertical. ORAS5 uses the Nucleus for European Modelling of the Ocean (NEMO v3.4.1) ocean coupled to the Louvain-la-Neuve Sea Ice Model (LIM2) sea-ice model. It includes a prognostic thermodynamic-dynamic sea-ice model with assimilation of sea-ice concentration data (Zuo et al., 2017, 2019). The ocean mixed layer is the layer immediately below the ocean air-sea interface and is typically tens of meters deep in summer while values of several hundred meters may be reached in winter. In this paper the interannually and monthly varying mixed layer depth (MLD) from ORAS5 is used to calculate the SST tendencies ($\Delta$SST) due to surface heat flux between 1989-2016. The ORAS5 MLD is the first depth at which the density difference, compared to density at 10 m depth, reaches $0.01\text{kg/m}^3$. The Ocean Reanalysis MLD agrees well with the observationally based MLD estimates in the mid-Atlantic (Toyoda et al., 2017). However, in deep convective regions, such as the Labrador Sea, the density difference MLD definition can overestimate MLD (Courtois et al., 2017).

## 2.3 Cyclone identification

Following Dacre et al. (2012) we identify and track the position of the 200 most intense cyclones in 20 years of the ERA-Interim dataset (1989-2009) during wintertime only (DJF) using the tracking algorithm of Hodges (1995). Tracks are identified using 6-hourly 850 hPa relative vorticity, truncated to T42 resolution to emphasize the synoptic scales. The 850 hPa relative vorticity features are filtered to remove stationary or short-lived features that are not associated with extratropical cyclones. The intensity of the cyclones is measured by the maximum T42 vorticity. The 200 most intense DJF cyclone tracks with maximum intensity in the North Atlantic ($70° − 10°$ W, $30° − 90°$ N) are used in this study and they account for $19\%$ of the total number of identified North Atlantic cyclone during this period. These tracks are shown in figure 1(a). The cyclones generally propagate in a north-easterly direction, from the east coast of America towards Iceland. The position of the cyclones 24 hours before maximum intensity (*max -24*) are predominantly over the Gulf Stream (figure 1(b)). By maximum intensity (*max*) the majority of cyclones are located east of Newfoundland (figure 1(c)). During the decaying stage of the cyclones evolution (*max + 24*) the cyclones are more uniformly distributed across the North Atlantic (figure 1(d)).

## 2.4 Cyclone-relative composites

The fields, described in section 2.1, are extracted from ERA-Interim at each of the 6 hourly locations of the cyclone within a $30°$ radius surrounding the cyclone centre. Cyclone-relative composites are produced by averaging over all cyclones. Following Catto et al. (2010), before compositing the fields are rotated according to the direction of travel of each cyclone such that the direction of travel in the composite becomes the same for all cyclones. Since the cyclones have quite different propagation directions, performing the rotation ensures that mesoscale features such as warm and cold fronts are approximately aligned and are not smoothed out by the compositing. As this method assumes that the cyclones all intensify and decay at the same rate

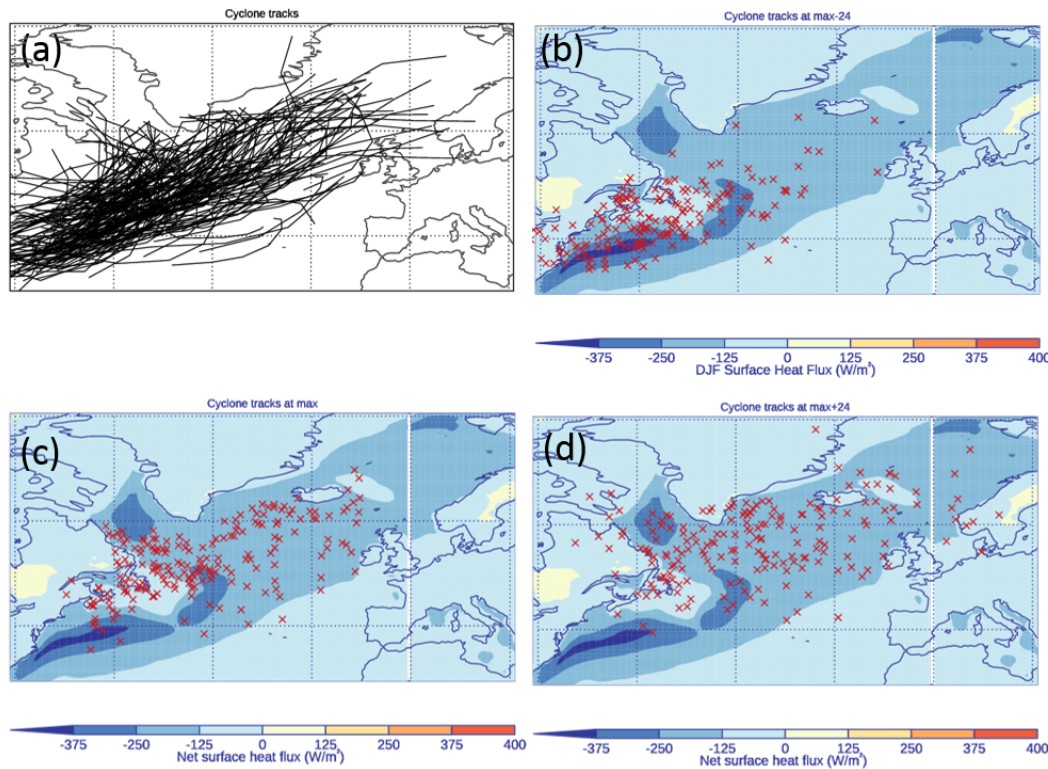

**Figure 1.** (a) Tracks of the 200 most intense DJF North-Atlantic storms between December 1989 and February 2009 (black). Position of the cyclones (b) 24 hours before maximum intensity (max-24), (c) at maximum intensity (max) and (d) 24 hours after maximum intensity (max+24) (red crosses). Overlaid on the DJF North Atlantic 1989-2015 net surface heat flux climatology, $Q_N$ (W/m$^2$).

only the 200 most intense cyclones are included in the composite. The 850 hPa relative vorticity mean and standard deviation of the of these cyclone are shown in Dacre et al. (2012) (their figure 2(c)). Limiting the number of cyclones produces a more homogeneous group in terms of their evolution but will bias the mean fields to be typical of the most intense cyclones. Data are only included in the composite where grid points lie over ocean surface.

### 2.5 Cyclone and environmental flow partition

To understand the part played by cyclones in the development of the cold anomaly during the winter of 2013/2014 the net surface heat flux is partitioned into environmental and cyclone components. To partition $Q_N$ into a part due to environmental flow and a part associated with cyclones we combine the cyclone tracks for that season with a masking method. A cyclone mask is calculated for each time step where the regions influenced by a cyclone are given a value of one (i.e., they are inside the cyclone mask) and regions that are not influenced are given a value of zero (i.e., they are outside the cyclone mask) (Hawcroft et al., 2012; Sinclair and Dacre, 2019). The heat flux associated with cyclones are asymmetric around the cyclone. Large

negative flux is found close to the cyclone centre and extending along the trailing cold front which lies to the west of the cyclone centre. To account for this, a cyclone mask at a given time, $t$, is created by identifying the position of a cyclone and also it's position during the previous 30 hours representing the cyclone's cold wake. A mask is created which extends $14°$ in a radial circle from each track point, creating an elongated oval shape which encompasses both the cyclone centre and the cold wake (figure 2). This is equivalent to taking a swath with circles around all cyclone positions in the previous 30 hours. The $Q_N$

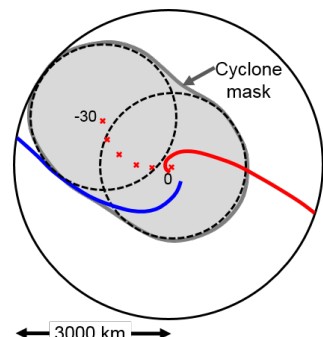

**Figure 2.** Schematic of cyclone masking method. Red and Blue lines show the approximate position of the cyclone warm and cold front respectively. Red crosses show the position of a given cyclone at time (t) and at 6-hourly intervals up until 30 hours previously (-30). The outer circle shows the 30 degree radius circle used to produce the composites in figures 5 to 8. The smaller dashed circles show example locations of a 14 degree radius circle centred on the cyclone position at t=0 and t=30. The grey shading shows the extent of the cyclone mask encompassing the cyclone's centre and cold wake.

anomaly when cyclones are not present is associated with the environmental flow. The $Q_N$ anomaly within the cyclone masks is associated with cyclones embedded within the environmental flow.

## 2.6    Sea Surface Temperature Tendency

In order to calculate the SST tendency ($\Delta SST$) we must take into account the mixed layer depth (MLD) through which the heating or cooling due to the surface fluxes is mixed ($h$). Mixing in the ocean is assumed to occur between the surface and the MLD obtained from ORAS5. Assuming a well-mixed layer, the SST tendency due to $Q_N$ and MLD ($\Delta SST_{Q_N}$) is given by;

$$\Delta\text{SST}_{Q_N} = \frac{1}{\rho c_p}\frac{Q_N}{h}, \tag{1}$$

where $\rho$ is the density of sea water, $1024 \text{kgm}^{-3}$; $c_p$ is the specific heat capacity of sea water, $4000 \text{Jkg}^{-1}\text{K}^{-1}$.

The SST tendency anomaly due to air-sea interactions (ASI), $\Delta\text{SST}'_{ASI}$, for the 2013/2014 season is determined by subtracting the climatological SST tendency from the 2013/2014 SST tendency and is given by;

$$\Delta\text{SST}'_{ASI} = \frac{Q_N^i - \overline{Q_N}}{\rho c_p \overline{h}} + \frac{Q_N^i}{\rho c_p}\left(\frac{1}{h^i} - \frac{1}{\overline{h}}\right) + \frac{Q_{ENT}^i - \overline{Q_{ENT}}}{\rho c_p \overline{h}}, \tag{2}$$

where the overbar represents the 1989-2015 climatological value. The SST tendency anomaly can be separated into the anomaly associated with (i) anomalous $Q_N$ (term 1 in equation 2), (ii) anomalous MLD (term 2 in equation 2) and (iii) anomalous

entrainment through the base of the mixed layer, $Q_{ENT}$ (term 3 in equation 2). Since we have no measurements of the

entrainment flux anomaly across the ocean boundary layer it is estimated to be 20% of the surface $Q_N$ anomaly (Stull, 1988). This estimate for the entrainment flux neglects contributions made by wind-driven turbulence.

## 3  Results

### 3.1  North Atlantic heat flux and SST tendency climatologies

Figure 3 shows the average heat flux for the period DJF 1989-2015 over the North Atlantic. The net surface solar radiation is

positive with a meridional gradient of $50\,\mathrm{Wm}^{-2}1000\mathrm{km}^{-1}$ (figure 3(a)) and the net thermal radiation (figure 3(b)) is negative with a magnitude between -50 and $-100\,\mathrm{Wm}^{-2}$. The sensible heat flux (figure 3(c)) is generally positive over land and negative over the ocean with negative flux between -50 and $-150\,\mathrm{Wm}^{-2}$ in the Gulf Stream and Davis Strait regions caused by the advection of cold air from the land over relatively warm oceans in DJF. The latent heat flux (figure 3(d)) is generally negative, with a band of enhanced negative flux extending in a north-eastwards direction from the east coast of the US towards Iceland

with the values $> 200\,\mathrm{Wm}^{-2}$ found in the west of the North Atlantic. The net heat flux, $Q_N$, (figure 3(e)), is negative over the majority of the domain. A combination of $Q_H$ and $Q_E$ results in maximum negative heat flux $> 300\,\mathrm{Wm}^{-2}$ in the Gulf Stream region. The largest $Q_N$ are co-located with the position of the North Atlantic storm track (figure 3(f)) which also exhibits a pronounced south-west to north-east tilt similar to the storm track in figure 1(a).

Figure 4(a) and (b) show the climatological SST and the average SST change over DJF ($\Delta\mathrm{SST}_{TOT}$) for the period 1989-

2015. During DJF the North Atlantic cools by an average of 2K (figure 4(b)). The cooling is greatest where the SST gradient is largest in the Gulf Stream region (dotted box in figure 4(a)), with cooling of over 6 K over the winter. Unlike the climatological $Q_N$ shown in figure 3(e) the region of highest $\Delta\mathrm{SST}_{TOT}$ does not extend over the North Atlantic towards Iceland but remains close to the east coast of the US.

In order to calculate the SST tendency due to $Q_N$ we must take into account the mixed layer depth (MLD) through which

heating or cooling at the surface is mixed using equation 1. Figure 4(c) shows the climatological MLD. The MLD ranges from a few 10s of metres close to Newfoundland, to over 1000m in the Labrador and Norwegian seas where deep convection occurs. On average the MLD deepens by 50% between December and February outside the deep convection regions (not shown). Taking the MLD into account restricts the largest $\Delta\mathrm{SST}_{Q_N}$ to the western North Atlantic region (figure 4(d)). The difference between $\Delta\mathrm{SST}_{TOT}$ and $\Delta\mathrm{SST}_{Q_N}$ is shown in figure 4(e). Differences $> 6K$ are found close to the coast where the western

boundary currents transport warm waters north resulting in reduced cooling over the DJF period than would be expected due to $Q_N$ alone.

### 3.2  Cyclone-relative heat flux composites

The north-south SST gradient in figure 4(a) is important for creating negative heat flux when cold dry air, from higher latitudes, is advected over relatively warm ocean surfaces. The largest heat flux occurs when large differences in temperature or moisture

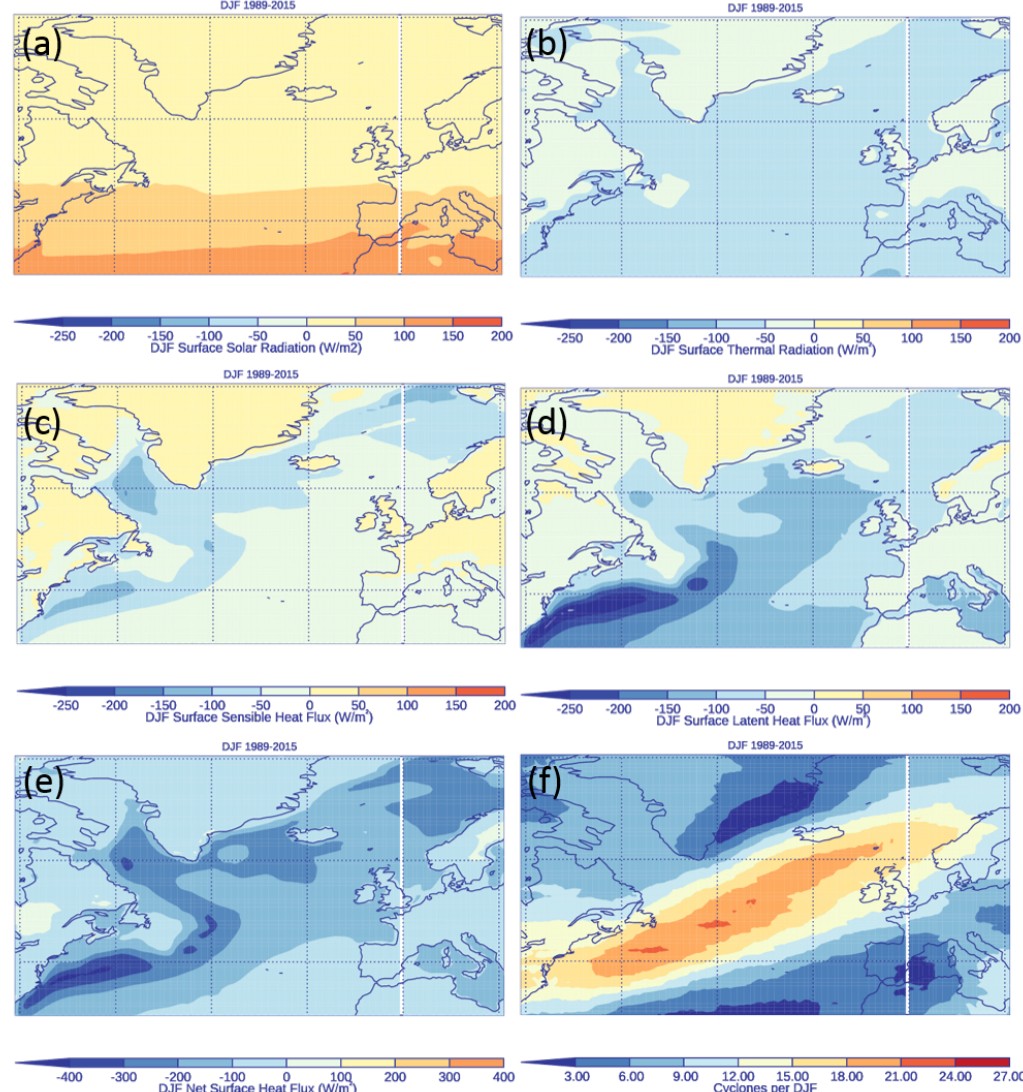

**Figure 3.** North Atlantic 1989-2015 heat flux climatologies (W/m$^2$). (a) Surface solar radiation $Q_{SW}$, (b) surface thermal radiation $Q_{LW}$, (c) surface sensible heat flux $Q_H$, (d) surface latent heat flux $Q_E$ and (e) net surface heat flux $Q_N$. Note change in scale in figure (e). Positive flux is into the surface and negative flux is into the atmosphere. (f) 1989-2015 climatological number of cyclones per DJF season within 12° of grid point.

between the surface and overlying atmosphere are co-located with enhanced wind speeds. Given the spatial distribution of SST's, atmospheric temperatures and moisture content, this is likely to occur when there are anomalously strong meridional winds such as found ahead of and behind extratropical cyclones. Thus, whilst the maximum climatological SST gradient controls where locally high surface flux occurs, the presence of extratropical cyclones are likely to control when the largest

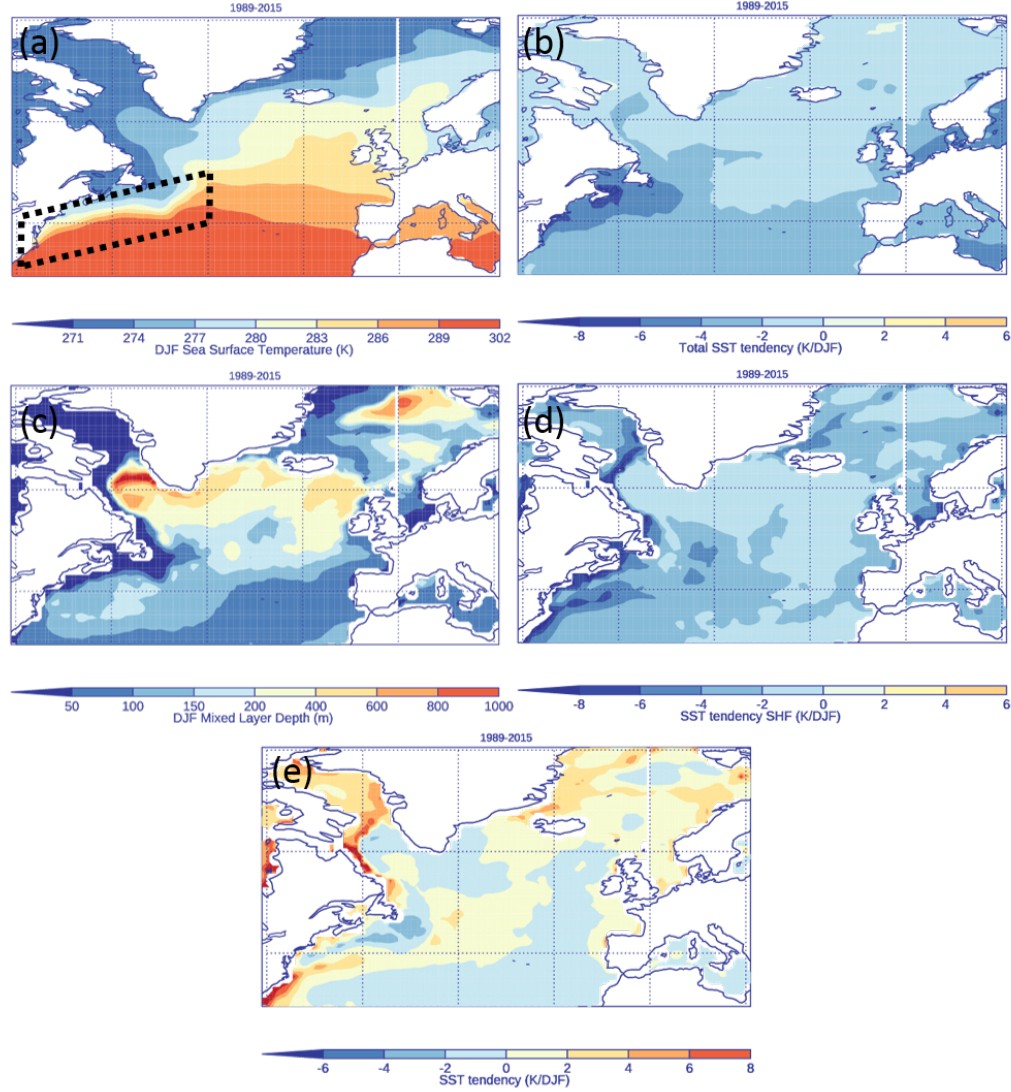

**Figure 4.** North Atlantic DJF 1989-2015 (a) sea surface temperature (SST, K), black dotted box outlines region of the Gulf Stream, (b) $\Delta \mathrm{SST}_{TOT}$ (K/DJF), (c) mixed layer depth (m), (d) $\Delta \mathrm{SST}_{Q_N}$ (K/DJF), (e) Difference between $\Delta \mathrm{SST}_{TOT}$ and $\Delta \mathrm{SST}_{Q_N}$.

heat flux occurs and its magnitude (Rudeva and Gulev, 2011; Ogawa and Spengler, 2019). In addition, the similarities between

the spatial pattern of climatological $Q_N$ (figure 3(e)) and cyclone numbers per season (figure 3(f)) suggests that cyclones play

a role in generating $Q_N$. In this section we investigate $Q_N$ in a cyclone-relative frame of reference using the methodology

described in sections 2.3 and 2.4.

Figure 5 shows composite cyclone centred heat flux for the 200 most intense cyclones occurring between 1989-2009. The

cyclones are at maximum intensity (maximum 850 hPa relative vorticity) when they are located at the centre of the domain

and have been rotated so that they all travel from left to right. $Q_{SW}$ (figure 5(a)) contribution to $Q_N$ is positive and, like the Earth-relative perspective (figure 3(a)), there is a weak gradient. However, because the cyclones are typically travelling in a north-eastwards direction and they have been rotated, this gradient is also rotated. $Q_{LW}$ (figure 5(b)) is negative everywhere and small, similar to the Earth-relative perspective. $Q_{LW}$ is slightly enhanced in the region behind the cold front, potentially due to a reduction of cloud, which reduces the downwelling radiation. $Q_H$ (figure 5(c)) shows a dipole structure at this stage

in the cyclones evolution, with negative flux behind the cyclone centre and positive flux in the cyclones' warm sector. $Q_E$ (figure 5(d)) is negative everywhere, a minimum in an extended region behind the cyclone and reduced ahead of the cyclone. $Q_N$ (figure 5(e)) is therefore negative surrounding the cyclone centre with a minimum behind the cyclone. Negative flux $> 200$ W/m$^2$ occurs within 1000 km of the cyclone centre but extend almost 2000 km behind the cyclone due to a combination of $Q_H$ and $Q_E$ occurring behind the cold front.

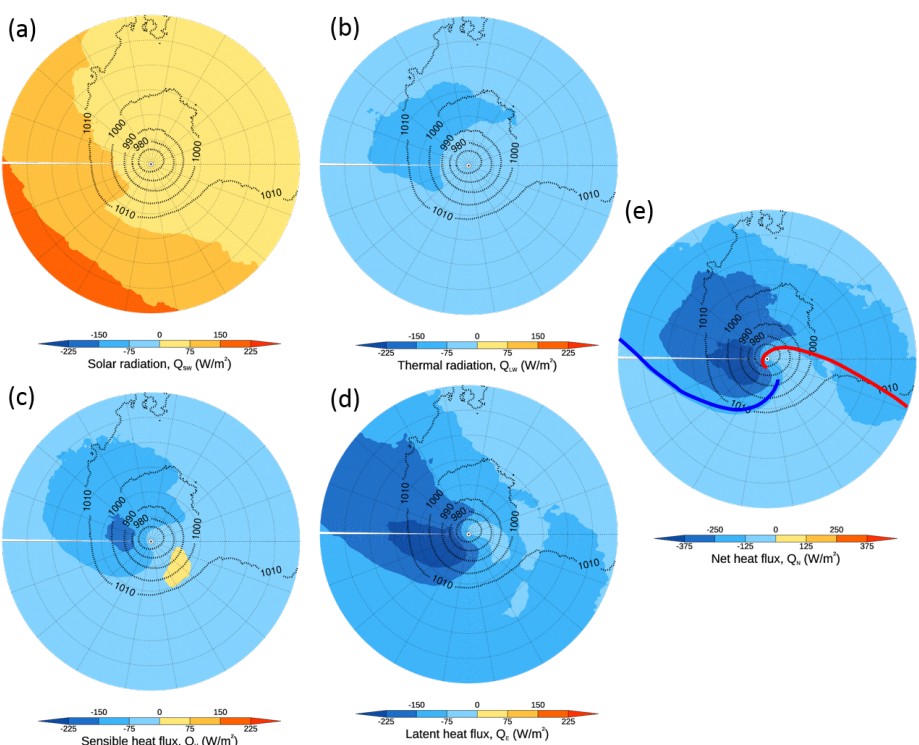

**Figure 5.** Composite cyclone centred heat fluxes (W/m$^2$) for cyclones at maximum intensity. (a) Surface solar radiation $Q_{SW}$, (b) surface thermal radiation $Q_{LW}$, (c) surface sensible heat flux $Q_H$, (d) surface latent heat flux $Q_E$ and (e) net surface heat flux $Q_N$ overlaid with mslp. Note the change in colour scale in figure (e). Positive fluxes are into the surface and negative fluxes are into the atmosphere. The radius of the circles is 3000 km and the cyclones are travelling from left to right. The blue and red lines in (e) represent the approximate positions of the cold and warm fronts at max intensity.

In order to illustrate the processes leading to negative $Q_H$ behind the cyclone cold front and positive $Q_H$ in the warm sector it is necessary to examine the temperature characteristics of the different airmasses in these regions. Figure 6(a) shows the composite 10 m air temperature overlaid with 925 hPa winds and figure 6(b) shows the SST. The 10 m air temperature exhibits a wave like structure whilst the SST gradient is more linear. This results in a large negative near surface temperature difference (SST $> 10$ m temperature) behind the cold front. Cyclonic winds advect relatively cold air over a warm ocean surface behind

the cyclone resulting in negative $Q_H$. Ahead of the cyclone cyclonic winds advect warm air over the ocean surface. 6 hours before maximum intensity the SST $< 10$m air temperature ahead of the cyclone (not shown) but at maximum intensity the difference is close to zero (white contour in figure 6(b) shows -0.4 K). Since $Q_H$ is a 6-hour average, this results in the positive $Q_H$ observed in figure 5(c). Figure 6(d) and (e) show the 10 m specific humidity and the saturation specific humidity at the SST respectively. The 10 m specific humidity also has a pronounced wave structure with drier air behind the cold front and moister

air in the warm sector. Behind the cold front the saturation deficit is $> 4$ g/kg (figure 6(f)) causing evaporation of moisture from the surface and large negative $Q_E$ observed in figure 5(d). Ahead of the cyclone the moisture deficit is $< 1$ g/kg significantly reducing the magnitude of $Q_E$.

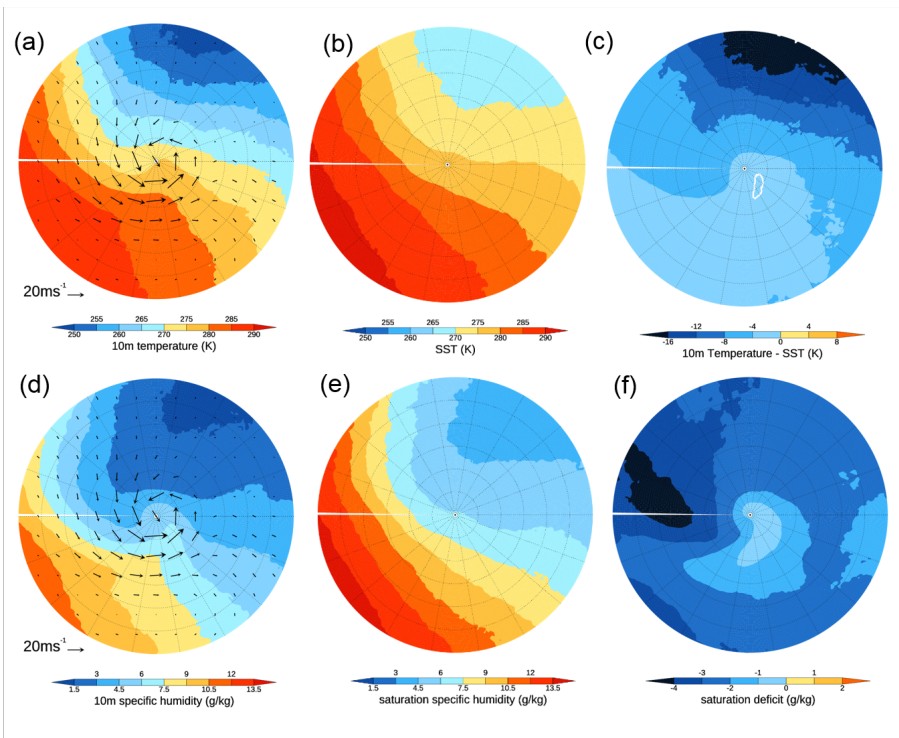

**Figure 6.** Cyclone centred fields at maximum intensity. (a) 10m air temperature (K), (b) SST (K), (c) 10m temperature - SST (white contour is -0.4 K) (K), (d) 10m specific humidity (g/kg), (e) saturation humidity of SST (g/kg), (f) 10m specific humidity - saturation humidity of SST (g/kg). (a) and (d) overlaid with 925hPa winds.

Figures 7(a)-(c) show composite $Q_N$ centred on cyclones at different stages in their evolution. As the cyclones start to intensify negative $Q_N$ behind the cold front strengthens (figure 7(b)). During the mature stages of the cyclone evolution

(figure 7(c)) $Q_N$ begins to decrease and to wrap cyclonically around the cyclone centre. This is due to the fact that the cold front typically rotates cyclonically towards the warm front as the cyclone reaches maturity. These results are consistent with the findings of Rudeva and Gulev (2011) who found that turbulent heat flux increases with cyclone intensity.

Interestingly, throughout the cyclone lifecycle there is a secondary minimum in $Q_N$ occurring approximately 2000 km ahead of the cyclone location. This secondary minima does not change magnitude so is unlikely to be affected by the cyclones at the

centre of the composite. It is possible that this second minima is due the presence of a downstream cyclone indicated by the composite mslp contours, which extend towards the upper-right quadrant of the domain.

Since many of the 200 cyclones contributing to the composite $Q_N$ (figure 5(e)) are generated over the Gulf-Stream region it is possible that large negative $Q_N$ occurring behind of the cyclone centre could be an artifact of their preferential cyclogenesis over a region of climatologically large negative $Q_N$ (figure 3(e)). To determine how $Q_N$ compares to the background values

we normalise $Q_N$ anomalies by subtracting the climatological field at the position of each cyclone and divide by the standard deviation of the climatology at the same location. Figures 7(d)-(f) show normalised $Q_N$ anomalies at different stages of the cyclone evolution. Negative anomalies indicate anomalously large heat flux into the atmosphere, with a value of -1 being 1 standard deviation larger than the climatological mean. Positive anomalies indicate anomalously small heat flux into the atmosphere, with a value of +1 being 1 standard deviation smaller than the climatological mean. At all stages of the cyclone

evolution negative $Q_N$ behind the cyclone centre is more than $0.5$ standard deviations greater than the mean for strong cyclones and is more than $1$ standard deviations greater than the mean at maximum intensity (figure 7(e)). Ahead of the cyclone $Q_N$ is $0.4 - 0.8$ standard deviations greater than the mean at maximum intensity (figure 7(e)) due to warm air advection in the warm sector of the cyclone. Note that towards the edges of the domain many of the gridpoints are over land so have been excluded, therefore the sample size contributing to the composites is small resulting in noisy field.

## 3.3 Cyclone-relative SST tendency

Using equation 1 the evolution of the cyclone-relative SST tendencies due to $Q_N$ can be estimated. Figure 8 shows $\Delta\text{SST}_{Q_N}$ per day for cyclones at different stages in their evolution. The patterns of $\Delta\text{SST}_{Q_N}$ are similar to the patterns of $Q_N$ (figure 7(a)-(c)) showing that the surface flux is the dominant variable in the cooling and that variations in mixed layer depth are less important. As for $Q_N$, $\Delta\text{SST}_{Q_N}$ increases as the cyclones reach maximum intensity with maximum SST tendencies of

0.2K/day occurring in the cold sector behind the cold front. The normalised $\Delta\text{SST}_{Q_N}$ anomalies are calculated by subtracting the climatological $\Delta\text{SST}_{TOT}$ and dividing by the standard deviation of $\Delta\text{SST}_{TOT}$ (figures 8(d)-(f)). $\Delta\text{SST}_{Q_N}$ is larger than the climatological mean behind the cold front. However in this case the intense cyclones only reduce $\Delta\text{SST}$ by up to 0.25 times the standard deviation.

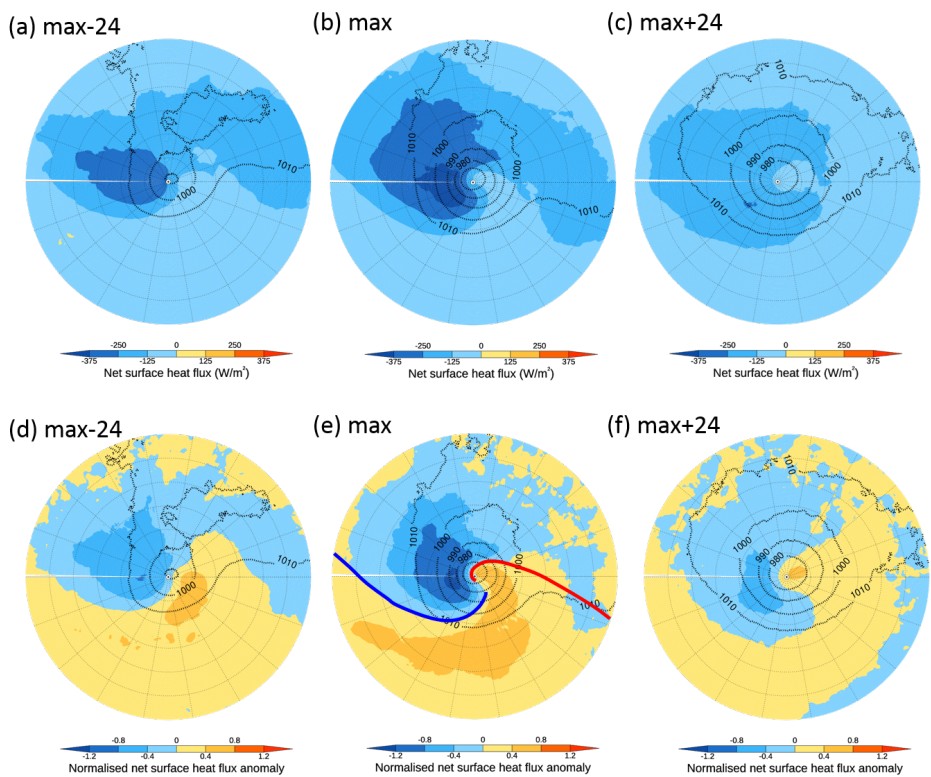

**Figure 7.** Evolution of cyclone centred (a)-(c) net surface heat flux $Q_N$ (filled contours, W/m$^2$), (d)-(f) normalised net heat flux anomaly (filled contours) and mslp (black contours, hPa). Cyclones reach the centre of the domain (a),(d) 24 hours before maximum intensity, (b),(e) at maximum intensity and (c),(f) 24 hours after maximum intensity. In (d)-(f) negative normalised heat flux anomalies indicate anomalously large heat flux into the atmosphere and positive anomalies indicate anomalously small heat flux into the atmosphere compared to climatology. The blue and red lines in (e) represent the approximate positions of the cold and warm fronts at max intensity.

## 4   2013/2014 heat flux anomalies

$\Delta\text{SST}_{Q_N}$ cooling associated with each individual extratropical cyclone is of the order $0.1-0.2$K/day (figures 8(a)-(c)) therefore if many cyclones track over the same location we might expect to see a signature of the storm track in the seasonal $Q_N$ and SST anomaly patterns. Figure 9(a) shows the tracks of cyclones in the North Atlantic region. Applying the cyclone masking methodology described in section 2.6 to the 2013/2014 winter cyclones we see that the cyclones track in a south-west to north-east direction in a narrow band that extends from the east coast of the US towards the UK (figure 9(b)). This season was
unusually stormy in the UK with cyclones passing over the UK every 3 days (Priestley et al., 2017). The effect of propagation speed is taken into account in the masking methodology since multiple timesteps for a single cyclone contribute to the seasonal mask fraction. We also apply the cyclone masking methodology to winter cyclones between 1989-2015 (figure 9(c)) and find that, in comparison to the 1989-2015 average, the 2013/2014 the storm track was more active, with a higher number of cyclones

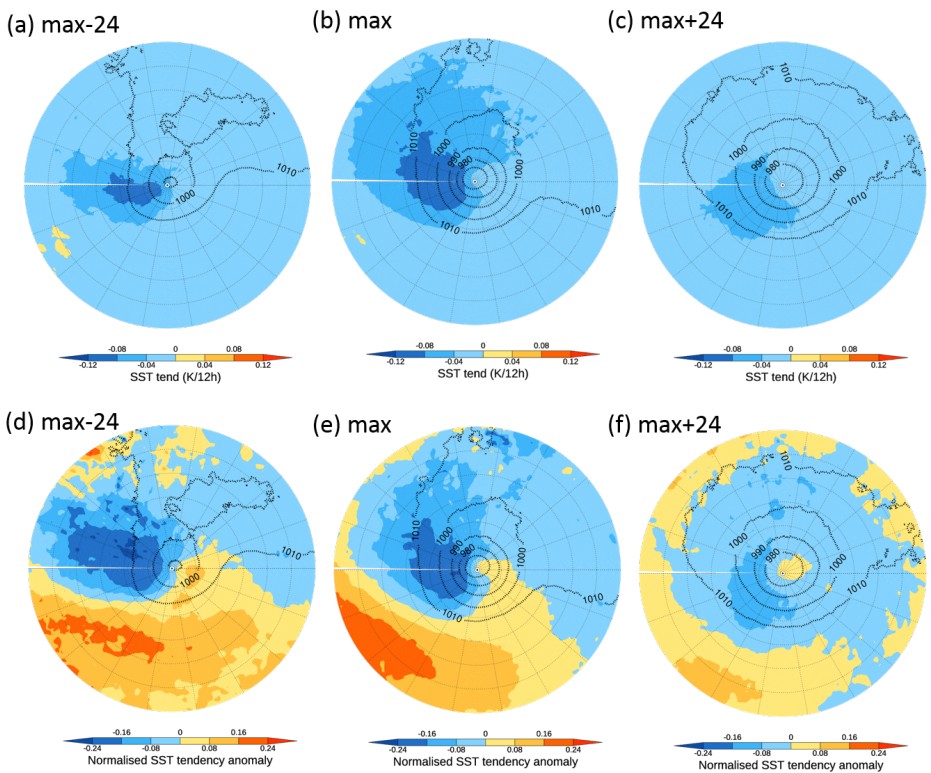

**Figure 8.** Evolution of cyclone centred (a)-(c) $\Delta SST_{Q_N}$ (filled contours, K/day), (d)-(f) Normalised $\Delta SST_{Q_N}$ anomaly (filled contours) and mslp (black contours, hPa). Cyclones reach the centre of the domain (a),(d) 24 hours before maximum intensity, (b),(e) at maximum intensity and (c),(f) 24 hours after maximum intensity. In (d)-(f) Negative normalised $\Delta SST_{Q_N}$ anomalies indicate anomalously large SST cooling due to $Q_N$ and positive anomalies indicate anomalously small SST cooling compared to climatology.

and also more zonal than usual (figure 9(d)). This was associated with an anomalously strong and zonally elongated upper-level
westerly jet in 2013/2014 (Kendon and McCarthy, 2015). More cyclones travelled towards western Europe than Iceland than usual.

    It was shown in figure 7 that intensifying cyclones create negative $Q_N$ behind the cold front, where cold dry air is advected over a warm ocean surface. Therefore we hypothesise that an anomalously strong and zonal storm track will result in an anomalously strong and zonally orientated seasonal $Q_N$ anomaly. The 2013/2014 DJF season $Q_N$ is shown in figure 9(e) and
the 2013/2014 $Q_N$ anomaly shown in figure 9(f). The seasonal $Q_N$ anomaly has a tripole pattern, with stronger negative $Q_N$ in the mid-North Atlantic and weaker negative $Q_N$ over the Gulf Stream and in the Norwegian and Greenland Seas. In the mid-Atlantic this is consistent with the shift in the storm track, with cyclones travelling more zonally from the US towards western Europe rather than north-eastwards towards Iceland.

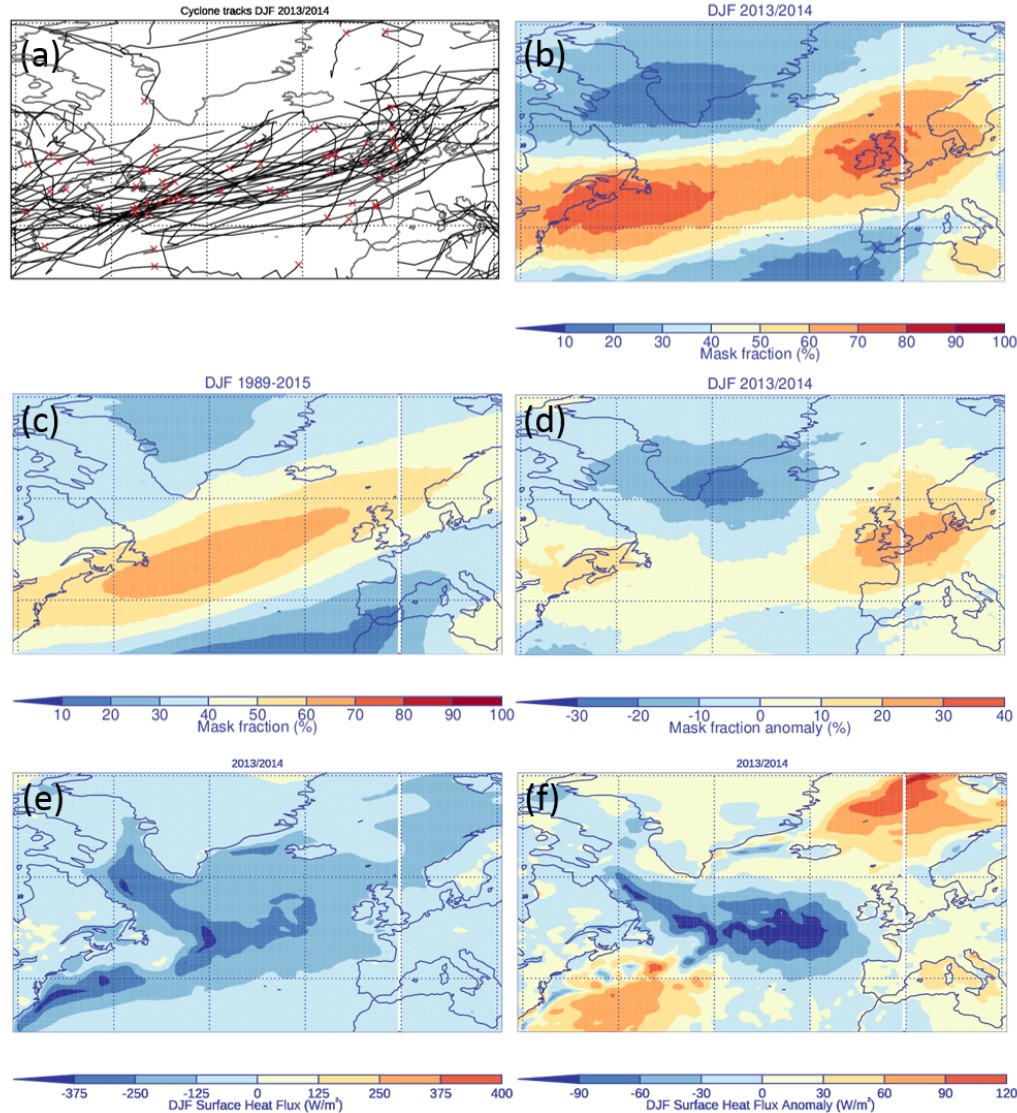

**Figure 9.** (a) 2013/2014 cyclone tracks (black) with position of maximum intensity (red crosses), (b) 2013/2014 fraction of time within cyclone mask, (c) 1989-2015 mask fraction, (d) anomalous mask fraction, (e) 2013/2014 $Q_N$ (W/m$^{-2}$) and (f) anomalous $Q_N$ (W/m$^2$).

Figures 10(a) and (b) show UK Met Office synoptic analysis charts at 00UTC on 24 and 20 December 2013 respectively.
At both times there is a low pressure centre situated to the west of Scotland and long trailing cold fronts extending across the north Atlantic. Figures 10(c) and (d) show $Q_N$ at the corresponding times and figures 10(e) and (f) show the cyclone masks. The red lines shows the full track of the cyclones and the elongated oval masks the location of the cyclones at the analysis time and 30 hours earlier. The mask captures $Q_N$ surrounding the cyclone centres and the cyclone's cold wake. Sensitivity tests have been performed using a radial circle ranging from 12-16° and masks extending 24 to 36 hours prior to the analysis time.

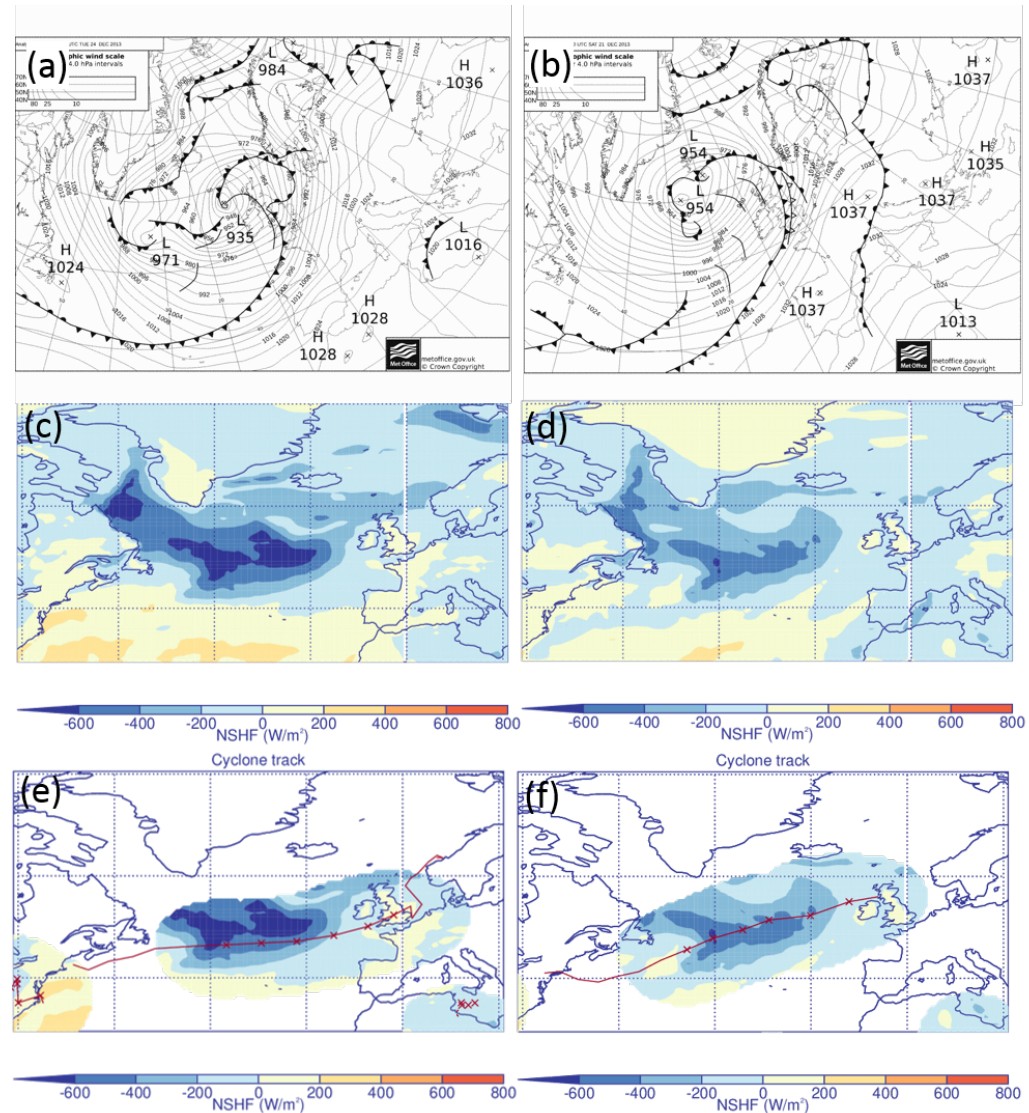

**Figure 10.** (a,b) UK Met Office synoptic analysis charts, (c,d) $Q_N$ (W/m$^2$), (e,f) cyclone mask overlaid with tracks for (a,c,e) 00 UTC 24 December 2013 and (b,d,f) 00 UTC 20 December 2013. Red crosses show the position of cyclones at the analysis time and 30 hours previously.

## 4.1   2013/2014 SST anomalies

Figure 11(a) shows the 2013/2014 DJF SST tendency anomaly ($\Delta$SST$'$) that is associated with anomalous $Q_N$. As expected $\Delta$SST$'$ due to anomalous $Q_N$ closely resembles the $Q_N$ anomaly (shown in figure 9(f)) with anomalous cooling in the mid-North Atlantic where the flux anomaly is negative, and anomalous warming (less cooling than climatology) in the Gulf Stream

and Norwegian sea regions. Small differences are due to spatial inhomogeneity in the North Atlantic climatological MLD.

Figure 11(b) shows the 2013/2014 DJF $\Delta SST^{'}$ that is associated with anomalous MLD. The 2013/2014 MLD is shallower than the climatological average over much of the domain, particularly near the Gulf Stream region, and deeper than climatology in the mid-Atlantic region. In the mid-North Atlantic the enhanced negative $Q_N$ results in negative buoyancy and thus mixing, deepening the MLD. Therefore, the surface flux decreases the temperature over a deeper layer of the ocean than usual, which reduces the direct SST cooling due to $Q_N$. At the same time, the increased MLD mixes colder water from below into the mixed

layer which cools the surface indirectly. Figure 11(c) shows the sum of the $\Delta SST^{'}$ due to anomalous $Q_N$, anomalous MLD and anomalous entrainment ($\Delta SST^{'}_{ASI}$). It shows the same tripole pattern as the $\Delta SST_{TOT}$ anomaly (figure 11(d)) which has an average SST cooling anomaly of -1.0K in the mid-North Atlantic region (black box in figure 11(d)). In the mid-North Atlantic region, the $\Delta SST^{'}_{ASI}$ accounts for $68\%$ of the total anomalous SST cooling in the mid-North Atlantic. The largest discrepancies occur along the east coast of North America suggesting that ocean dynamics are responsible for transporting

warmer/colder water into these regions via the western boundary currents. For example, off the coast of Nova Scotia the SSTs are very cold since the Labrador current flows south from the Arctic Ocean. Advection of relatively warm air over these cold SSTs in a region where the mixed layer depth is shallow results in a positive SST tendency anomaly.

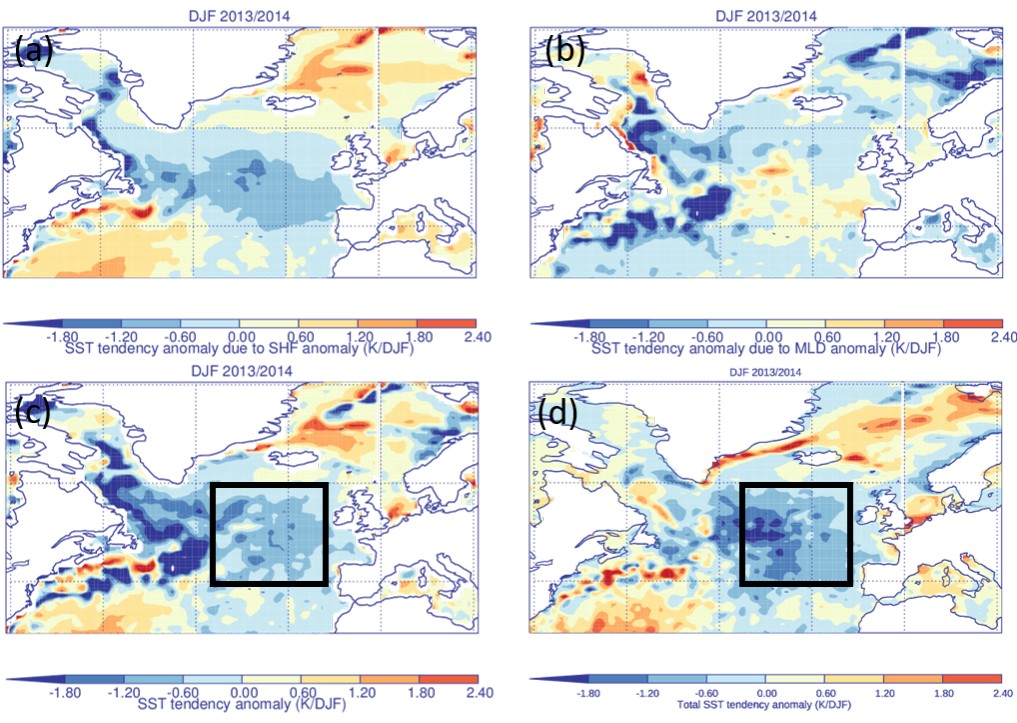

**Figure 11.** Anomalous $\Delta SST$ due to 2013/2014 (a) $Q_N$ anomaly, (b) MLD anomaly and (c) $\Delta SST_{ASI}$ anomaly. (d) $\Delta SST_{TOT}$ anomaly.

## 4.2 2013/2014 cyclone and environmental flow SST anomalies

In this section we partition the $Q_N$ anomaly into a part associated with the environmental flow (i.e., outside the cyclone masks described in section 2.6) and a part associated with the presence of cyclones (inside the cyclone masks described in section 2.6). Figure 12(a) shows the contribution to the DJF $Q_N$ anomaly due to both cyclones and the environmental flow and figure 12(b) shows the contribution that is due to the environmental flow only. Both figures 12(a) and (b) show a tripole pattern with anomalously negative heat flux in between $40 - 60°$N and anomalously positive heat flux over the Gulf Stream and Norwegian Seas. This suggests that the overall pattern is controlled by the environmental flow. The anomalously positive heat flux in the Norwegian Sea and Gulf Stream region may be due to a reduced number of cold air outbreaks (Papritz and Spengler, 2017; Papritz and Grams, 2018; Parfitt et al., 2016). In the mid-north Atlantic the negative $Q_N$ anomaly is enhanced, compared to contribution made by the environment (doubled from $32\%$ to $68\%$ of the total seasonal $Q_N$ anomaly), when cyclones are present. In the Norwegian sea and Mediterranean sea the positive $Q_N$ anomaly due to the environmental flow is suppressed by the presence of cyclones. Thus cyclones embedded within the environmental flow pattern tend to increase the negative surface heat flux, consistent with the cyclone-relative results in section 3.2. Varying the size of the cyclone mask radius from $12 - 16°$ results in a contribution to the total $Q_N$ anomaly when cyclones are present in the mid-North Atlantic that ranges from $62 - 72\%$ and varying the length of the cyclone mask from $24 - 36$ hours results in a cyclone contribution from $62 - 71\%$. This suggests that the conclusion that cyclones enhance the $Q_N$ anomaly does not depend strongly on the choices used to define the cyclone mask.

Figure 12(c) shows the contribution to the 2013/2014 $\Delta SST'_{ASI}$ due to both cyclones and the environmental flow and figure 12(d) shows the contribution that is due to the environmental flow only. As for the $Q_N$ anomaly, both show a tripole pattern suggesting that the environmental flow controls the anomalously large cooling in the mid-North Atlantic and the anomalously small cooling in the Norwegian Sea and Gulf Stream regions. In the mid-North Atlantic region, the negative $\Delta SST'_{ASI}$ is also enhanced (from $28\%$ to $41\%$ of the total seasonal $\Delta SST'$) when cyclones are present. This is less than their added contribution to the 2013/2014 $Q_N$ anomaly because the enhanced negative $Q_N$ is cooling a deeper layer of the ocean than usual which reduces the direct SST cooling.

## 5 Conclusions

In this paper we investigate both the SST cooling associated with individual cyclones and the SST cooling associated with the passage of multiple cyclones over the same location in the 2013/2014 season to determine how significant cyclones were in contributing to the anomalously large cooling that occurred during the 2013/2014 winter. We find that enhanced air-sea exchange of heat and moisture in the cold sector behind the cold front of an extratropical cyclone can lead to cooling of up to 0.2 K/day for the strongest cyclones. This cooling is relatively small compared to the variability in SST tendency in the North Atlantic, thus the 'cold wake' associated with the passage of an individual extratropical cyclone is difficult to observe in the instantaneous SST field.

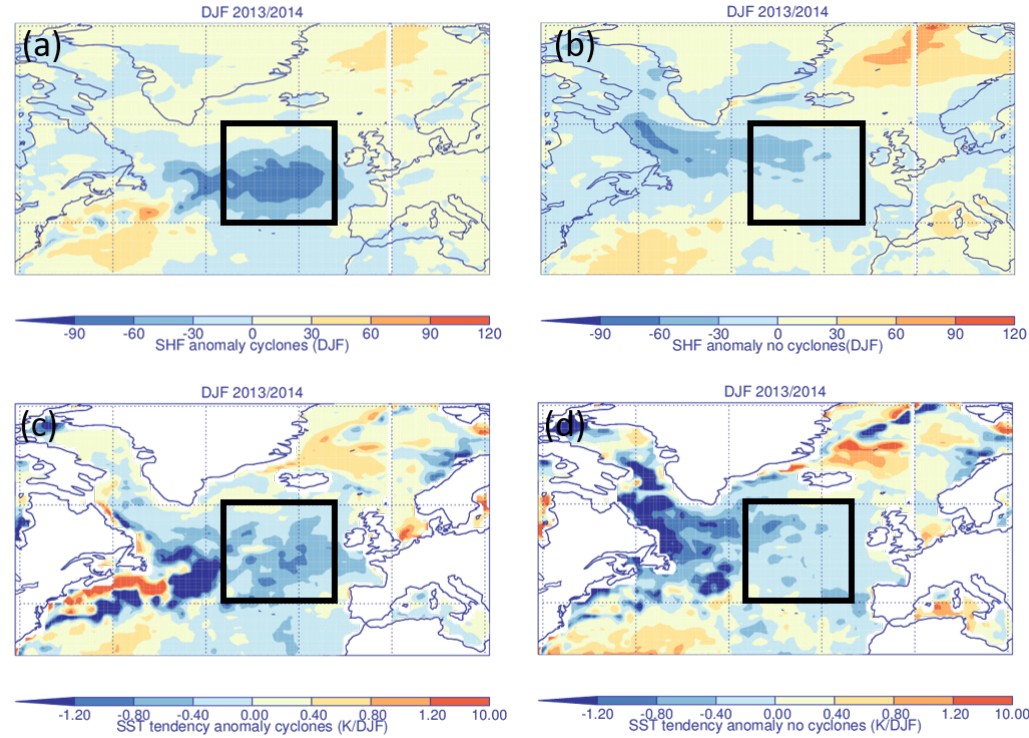

**Figure 12.** 2013/2014 anomalous $Q_N$ associated with (a) both cyclones and the environmental flow and (b) the environmental flow. 2013/2014 anomalous $\Delta SST$ due to anomalous $Q_N$, MLD and entrainment associated with (c) both cyclones and the environmental flow and (d) the environmental flow.

During the 2013/2014 DJF season there was a zonal band of anomalously large negative heat fluxes extending from the east coast of the US towards Europe. The mixed layer depth was also anomalously deep in the mid-North Atlantic due to enhanced mixing and entrainment of water into the mixed layer from below. The combination of these air-sea interactions accounts for $68\%$ of the total $\Delta SST$ anomaly. Thus air-sea interactions were very important in determining the anomalous SST cooling between December 2013 and February 2014.

In 2013/2014 the environmental flow pattern was anomalously zonal compared to climatology over the North Atlantic. This resulted in anomalously negative heat flux between $40-60°$N and anomalously positive heat flux over the Norwegian and Gulf Stream regions. When cyclones were present, heat flux from the ocean to the atmosphere was doubled in the mid-North Atlantic region. The enhancement of the negative heat flux caused a direct cooling of the ocean but also led to increased entrainment and thus a deeper mixed layer. This deepening reduces the overall effect of the cooling as the heat loss acts on a greater volume of water than normal. Consequently, while the SST tendency anomaly in the mid-North Atlantic was enhanced by the presence of cyclones it was by a smaller amount than might be expected due to a doubling of the heat flux. We conclude that both the

environmental flow and extratropical cyclones embedded within this flow played important roles in determining the extreme 2013/2014 winter season SST cooling.

*Author contributions.* H.F.Dacre performed the data analysis for this publication. S.A.Josey and A.L.M.Grant contributed to the scientific interpretation of the analysis.

*Competing interests.* The authors declare that they have no conflict of interest.

*Acknowledgements.* The ERA-Interim data were obtained freely from http://apps.ecmwf.int/datasets/. The ORAS5 data were obtained freely from the Integrated Climate Data Center at University of Hamburg http://icdc.cen.uni-hamburg.de. Information on how to obtain the cyclone identification and tracking algorithm can be found from http://www.nerc-essc.ac.uk/ kih/TRACK/Track.html and access obtained by emailing Kevin Hodges (k.i.hodges@reading.ac.uk). We thank Kevin Hodges for providing his ETC tracking code. SJ receives funding from the UK Natural Environment Research Council including the ACSIS and EMERGENCE programmes.

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
