# Peer review of "Extratropical cyclone induced sea surface temperature anomalies in the 2013/14 winter"

_Weather and Climate Dynamics, 2019_

## Referee Comment (RC1) · Anonymous Referee #1 · 3 Oct 2019

The paper documents the sea surface cooling by extratropical cyclones and its impact on the 2013/2014 winter SST in the mid North Atlantic. The conclusions and interpretations are adequately supported for the most part. The paper is well written and conclusions are concise and clear.

Specific comments

1. Does the warming tendency in the warm sector has any effect on SST? In Section 4.1, the cyclone mask is creased so as to encompass the cold front and the cyclone center. Does this method include the warm sector properly?

2. The authors focus on the 2013/2014 winter, but I expect that cyclones could play an important role even in other years. The authors might want to estimate cyclones

contribution to the winter climatology of the net heat flux using your cyclone masking technique. It would develop a much deeper understanding of the cyclones role.

3. In addition to the strength and number of cyclones, the propagation speed is probably also important for the cooling. The high fraction of time of cyclone mask in 2013/2014 around the UK seems to be partly due to the stagnation of cyclones (Fig. 8).

4. Is the anomalously zonal storm track in 2013/2014 associated with the westerly jet?

5. The distribution of the Qn anomaly in Figure 8f is different and shifted from that of the cyclones in Figure 8d. Why are they different?

6. The anomalous Qn not associated with cyclones in Figure 11b still has a tripole pattern. So do you think that the tripole pattern has basically nothing to do with cyclones?

7. L153-164. It is difficult to identify the position of the cold front and warm section in Figure 4. How about plotting the cold and warm fronts? These fronts could be delineated based on Figure 5 or the map of relative vorticity of wind.

Technical corrections

L38. of the wind driven currents

L128. over 6 K over the winter

L135. The density of sea water 1000 kg/m^3 might be acceptable, but the more practical value (like 1024 kg/m^3) should be used.

L143. figure 3(a)?

Figure 4. What do contour lines show?

L246. the conclusion does not

L250. the anomalous Qn

L250. figure 8(f)
* * *

---

## Referee Comment (RC2) · Anonymous Referee #2 · 11 Oct 2019

The paper explores a connection between SST anomalies and atmospheric cyclones in the North Atlantic. The paper is concise and well written. My major concern is that the authors used a cyclone dataset, while their main finding relates more to cold fronts that are possibly associated with cyclones. I suggest adding a dataset on the location of fronts and calculating anomalies behind objectively identified cold fronts (perhaps within a cyclone area or independently) rather than deducing the location of fronts within cyclones.

Specific comments:

Why only 2013/14 season is taken into account to calculate cumulative effect of the passage of multiple cyclones.There must be some other anomalous seasons.

[Figure]

Tilinina et al. 2018 (https://doi.org/10.1175/MWR-D-17-0291.1) investigate anomalously high heat fluxes in the North Atlantic during winter and related those to the cyclone activity. They concluded that the area of interaction between cyclones and anticyclones is very important for a heat flux anomaly. I wonder if this is also true for the summer season and it will also be nice to see some analysis on this.

l. 3: are the processes not fully understood or not quantified?

l.29: I believe it should be Rudeva and Gulev (2011)

5. l33: should it be left-rear quadrant?

l.41: I'd add 'ocean' surface mixed layer

l.89-92: you say 'MLD is the depth at which the density difference . . . . reaches 0.01 kg/m3' and then 'the density difference MLD can overestimate MLD'. Define MLD otherwise then.

l.98-100: 200 most intense cyclones - how does that number compare with the total number of cyclones for 1989-2009? How intense are those cyclones (perhaps, add a pdf intensity for all cyclones and those 200). As you focus on the North Atlantic, I'd suggest 30-70N, instead of 90N (though looking at the track in fig. 1 it will hardly make a difference for the results). Consider showing this area in Fig.1.

l.105-113: How composites are built should be better described here. It is only in sec. 4.1 that we find out that the radius of composites is 30deg (it is also mentioned in fig 4 caption). I believe that the rotation of composites does not help interpretation of the results as meridional gradients in some plots get also rotated (e.g., fig. 4), I'd recommend skipping this step.

l.111: Following your comment on the rate of intensification and decay, I think a pdf will be helpful (together with a pdf of intensity mentioned in my earlier comment)

l.116: give a range for the meridional gradient

All Figures: show lon/lat

Figures 1 and 2: Add Qsw,Qlw, etc. to the captions (as in other figures)

l.143: check the figure number

l.152: SST tendencies are discussed in the next section

l. 163: 'westward direction': as the composites are rotated it is hard to say where the west is.

L 159:164: how much are sensible and latent heat fluxes in summer different to those in winter in previous studies (in Tilinina et al. 2018 and Rudeva and Gulev 2011)? From this you may possibly deduce a potential effect of cyclones on SST in winter (which can also be estimated directly in another paper)

l.165: this sentence suggests that wind should also be shown in fig 5b

Fig 3b and d: fix colours in the colour scale (blue - negative, yellow/red - positive)

Fig.4: I'd comment that positive values are into the surface in the caption.

Fig.5: add 'air' to panel (a) caption

l.225-232: this paragraph should be in Methods

Fig.9: The relative sizes of the circles are wrong: if the big circle is 30 deg, as the small circle has a 14 deg radius.

l.237: I do not get why the mask shows the cyclone along the trailing cold front. It suggests that cold fronts always extend along the cyclone centres in the last 30 hours. If that is your assumption, that needs to be proved. As I said at the beginning of the review, I think you need objectively identified cold fronts instead of what has been invented here.

Fig 10: Maybe swap the panels to have 24 Dec on the right and 20 Dec on the left

l.245: 14-18% - what variable does it relate to?

l249, 250 : fig 11c and 8f, respectively

Fig11: perhaps I missed it, but was QN due to cyclones calculated for all cyclones in 2013/14, or the strongest? Fig. 11c shows SSTQN due to cyclones or any Qn? I think 11c should be due to cyclones only. Can you explain why strong negative anomalies in the west of the North Atlantic (fig. 11c) are not seen in fig. 11d? I'd say that 11d matches well with 11a, which makes sense, but anomalies in the west North Atlantic in 11d are confusing.

l.255: is it entrainment of the cold air?

l.265: As the mask stretches backwards from the cyclone centre, it captures the cold sector. However, the effect of the warm sector remains not assessed (which can also be done if warm fronts are identified).

Fig. 7 suggests that the warm sector will have relatively small effect during the max development, but at other stages of cyclone lifecycles the balance might be different.

Typos and language concerns:

The word 'flux' is often used in plural form (e.g., flux occur). My preference is either to say 'flux occurs' or 'fluxes occur'.

l. 73: magnitudes

l.101: position is

l.126, 166,173: Figure shows

l.128: 'teh' to 'the'

l.135: put comma after 4000Jkg-1K-1

l. 137: change 10's to 10s

l.147: remove 'are'

Fig 7: 'Normalised' and 'negative' should start with a small letter

l.246: remove 'is'

---

## Referee Comment (RC3) · Anonymous Referee #3 · 18 Oct 2019

Review of WCD-2019-6:

"Extratropical cyclone induced sea surface temperature anomalies in the 2013/14 winter"
by
Helen F. Dacre, Simon A. Josey, and Alan L. M. Grant

**Recommendation: Major revisions**

**General Comments:**

The Authors present an interesting analysis using ERA-Interim data to address the question how extratropical cyclones influence the SST in the Atlantic. They showcase one particular year that featured a significant SST anomaly and try to attribute a large fraction of this anomaly to anomalous cyclone activity in the same winter.

The manuscript is well written and the figures are clear, though the panel labels are sometimes difficult to see as they are on top of shaded figures. Overall, the paper presents a valuable contribution to the field and employs a novel diagnostic to attribute the surface fluxes to individual cyclones. However, there are several points in the paper that need further clarification, which are indicated in the comments below.

The mixed layer calculation has a caveat, because the authors assume that the depth has no variations throughout the year when they make seasonal budgets. One particular issue with that is that as the mixed layer depth changes, the sea state properties, in particular the stratification below the mixed layer, become important when the mixed layer depth increases. The actual heat content in the mixed layer will depend on the sea state below the mixed layer as well when net surface fluxes cause mixing. The entrainment of sea water below the column would need to be considered when the fluxes imply a net change in mixed layer depth. It would thus be interesting if the authors also show the seasonal tendency of the mixed layer depth in figure 3, not only the tendency in SST. Given the actual change of mixed layer depth together with the ocean stratification below the mixed layer could yield an estimate of the entrained energy into the changed mixed layer from below. This additional term in the heat budget could be accounted for and contrasted with the net surface forcing of the SST tendency.

Regarding the methodology of cyclone frequency, it is not clear if every cyclone is counted multiple times for the track densities of if some kind of anti-aliasing was employed. This would also influence how storm track activity is defined, as fewer but slower moving storms would yield a higher storm activity in terms of cyclone density compared to the same number of cyclones in a season with higher phase velocity. It would be great if the authors could further clarify how the cyclone track densities were calculated and how exactly one can thus understand an increased activity of cyclones. It would also be of interest if there were more extreme cyclones that particular year of interest, especially as the authors limit their analysis to the more intense systems.

A large fraction of the fluxes in the Gulf Stream region are associated with cold air outbreaks, of which a significant fraction is not necessarily associated with cyclones in the storm track region. Could the reduced QN fluxes in 2013/2014 south of the Gulf Stream region as well as in the Nordic Seas be thus actually associated with a reduced number of cold air outbreaks? For the Nordic seas, which also feature a significant anomaly in the presented analysis, Papritz and Spengler (J. Clim., 2017) showed that cold

air outbreaks account for the larges fraction of the surface fluxes in this region. Thus, the apparent anomalies are most likely mainly attributable to variations in cold air outbreaks and maybe only indirectly or in a reduced way associated with extratropical cyclones. Papritz and Grams (GRL, 2018) investigated the weather regimes associated with cold air outbreaks in the region of interest in the manuscript at hand. It would be interesting to put their findings and the given role of cold air outbreaks on the surface fluxes in the region in context with the presented findings.

In addition to cold air outbreaks, the role of cold fronts for surface fluxes in the Gulf Stream region has also been discussed recently, e.g., Parfitt and Czaja (2016) and other recent studies by the first author. It would be great if the authors could provide further context of the presented work to these studies.

The method to define the QN with the cyclone masks is not clear enough. It is difficult to follow what is actually summed up. At each time t for a given cyclone, the position of the cyclone and the preceding 30 hours positions are used, but is this done for every timestep in the cyclone evolution? How would this differ to just taking the swath with circles around all cyclone positions along the entire cyclone track? It would be great if the authors could provide further details about the employed method.

**Specific Comments:**
Page and line numbers refer to the ones on the manuscript.

P1 L7: The connection between the "cold wake" and "climatological variability" is not quite clear in this sentence. How is the size of the cold wake associated to climatological variability?

P1 L21: The argument about the role of cold fronts has also been discussed more recently, e.g., Parfitt and Czaja (2016) and other recent studies by the first author. What is the context of the presented work to these studies?

P2 L29: After citing the study by Zolina and Gulev (2003), the reader is a bit confused about the thus far identified fluxes associated with extratropical cyclones. If there is a controversy, it would be great if the authors could further highlight these conflicting results and possibly indicate as to why they are conflicting or if they will address these contrasting results.

P2 L28" …of the wind driven…

P2 L44: The authors comment on the role of ocean dynamics in the western Pacific, where oceanic advection probably plays a dominant role. However, the reader is left wondering if not similar arguments would also apply to the western Atlantic, the focus of this study, where strong oceanic currents are present. Are there no studies quantifying the role of oceanic anomalies in the western Atlantic? Good if the authors can also comment on the region of their interest in this context.

P2 L51: Another, more direct, connection between cold air outbreaks, cyclones, and the low-level baroclinicity in the western Atlantic is provided by Papritz and Spengler (2015) as well as Vanniere et al. (QJ, 2017).

P5 L128: "the winter"

P7 L140: See general comment about change of mixed layer depth throughout season. Some additional discussion about the influence of mixing and entrainment in the ocean would be valuable.

P7 L144: "heat fluxes occur"

P7 L147-149: This is also the argument of a recent study by Ogawa and Spengler (2019), who also emphasized the role of synoptic eddies on the climatological fluxed in the mid and higher latitudes.

P9 L183: "the cyclone lifecycle"

P11 L203: "the surface flux"

P12 L216: It is not necessarily obvious from the referenced figures that the storm track was more active, see general comment on cyclone track densities.

P13 L223: It is difficult to see how the QN anomaly and the storm track anomaly is "consistent". There appear to be more cyclones detected over the Gulf Stream region in the anomalous winter, though the net negative QN fluxes in this region appear to be reduced when compared to climatology. How can this be reconciled with the previous findings of the cyclone relative QN fluxes and SST changes?

P14 L225 and following: The methodology is not quite clear, see also general comments.

Fig. 9 caption: "red crosses show"

P15 L246: "conclusion does not"

P16 L250: It is not clear that the results indicated in this paragraph consider the data based on the cyclone swaths from the previous section.

P16 L254: The actual percentage of the SST difference cannot be really directly contributed to the fluxes, as it is a mix of local fluxes and advection, as well as entrainment from below that caused the total change. There might be compensating effects that cannot be accounted for in such a crude attribution without actually calculating a full budget considering all tendency terms.

P17 L262: Can the authors comment further on the relative contributions of potential other effects that make the attribution to individual cyclones difficult?

P17 L266: The statement about "higher than average cooling" appears to be rather regionally confined and there were also larger areas where this particular season featured reduced air-sea heat exchange. The authors should comment on this complex structure and put it in context to the observed cyclone distribution. Especially the western Atlantic area with reduced fluxes appears difficult to explain given the increased number of cyclones (Fig. 8f, 11d).

---

## Author Comment (AC2) · 20 Dec 2019

H. F. Dacre, S. A. Josey, A. L. M. Grant

December 20, 2019

**Reply to reviewer 2**

We would like to thank the reviewer for their comments on the paper. Below the reviewers comments are in black and the responses in blue italics. Changes to the paper are shown in red in the revised paper.

**General comment**

The paper explores a connection between SST anomalies and atmospheric cyclones in the North Atlantic. The paper is concise and well written.

*Thank you.*

My major concern is that the authors used a cyclone dataset, while their main finding relates more to cold fronts that are possibly associated with cyclones. I suggest adding a dataset on the location of fronts and calculating anomalies behind objectively identified cold fronts (perhaps within a cyclone area or independently) rather than deducing the location of fronts within cyclones.

*See response to specific point 22.*

**Specific comments**

1. Why only 2013/14 season is taken into account to calculate cumulative effect of the passage of multiple cyclones.There must be some other anomalous seasons.Tilinina et al. 2018 (https://doi.org/10.1175/MWR-D-17-0291.1) investigate anomalously high heat fluxes in the North Atlantic during winter and related those to the cyclone activity. They concluded that the area of interaction between cyclones and anticyclones is very important for a heat flux anomaly. I wonder if this is also true for the summer season and it will also be nice to see some analysis on this.

   *This is an excellent suggestion and one that was also made by reviewer 1. We have started to apply our cyclone masking technique to other years and seasons. However, including this analysis would increase the length of the paper significantly. Therefore, we will publish this work as a separate publication to avoid a very long paper.*

2. l. 3: are the processes not fully understood or not quantified?

   *'Quantified' is probably more correct so we have changed the wording in the abstract.*

3. l.29: I believe it should be Rudeva and Gulev (2011)

   *We agree, this citation has been changed.*

4. 5. l33: should it be left-rear quadrant?

*The SST cooling can be in either the right or left-rear part of the cyclone depending on the cold-front orientation. Therefore, this has been changed to 'rear part of the cyclone'.*

5. l.41: I'd add 'ocean' surface mixed layer

*Changed.*

6. l.89-92: you say 'MLD is the depth at which the density difference . . .. reaches 0.01 kg/m3' and then 'the density difference MLD can overestimate MLD'. Define MLD otherwise then.

*We have clarified that the overestimation of MLD using the density difference definition occurs predominantly in the deep convective regions and not over the entire North Atlantic domain. Since we focus on the mid-North Atlantic this does not influence our results.*

7. l.98-100: 200 most intense cyclones - how does that number compare with the total number of cyclones for 1989-2009? How intense are those cyclones (perhaps, add a pdf intensity for all cyclones and those 200). As you focus on the North Atlantic, I'd suggest 30-70N, instead of 90N (though looking at the track in fig. 1 it will hardly make a difference for the results). Consider showing this area in Fig.1.

*Between DJF 1989/1990 and DJF 2008/2009 there were 1050 cyclones identified with their maximum intensity in the North Atlantic domain. The top 200 cyclones represent the top 19% of the entire North Atlantic distribution as shown in figure 1. We have decided not to include this figure, but have stated the percentage of the total cyclones in the text. As the reviewer states, the tracks of the most intense cyclones will not change if the North Atlantic domain is reduced along it's northern boundary, therefore we have not reanalysed the data.*

8. l.105-113: How composites are built should be better described here. It is only in sec. 4.1 that we find out that the radius of composites is 30deg (it is also mentioned in fig 4 caption). I believe that the rotation of composites does not help interpretation of the results as meridional gradients in some plots get also rotated (e.g., fig. 4), I'd recommend skipping this step.

*We think that the reviewer must have missed this information, as it is stated in the first sentence of section 2.4 in which we describe the cyclone-relative compositing. Performing the rotation ensures that mesoscale features such as warm and cold fronts are approximately aligned and are not smoothed out by the compositing. We agree that this will also rotate meridional gradients, but feel that it is important to align the features we are interested in otherwise the composites become washed out. Therefore we have retained this step of the analysis.*

9. l.111: Following your comment on the rate of intensification and decay, I think a pdf will be helpful (together with a pdf of intensity mentioned in my earlier comment)

*Dacre et al. (2012) (their figure 2(c)) shows the mean intensification and decay rates of the top 200 cyclones as well as the spread around the mean. As the pdf has already*

[Figure]

Figure 1: Maximum relative vorticity reached by all 1050 north Atlantic cyclones. The grey shading represents the part of the distribution that includes the 200 most intense cyclones.

*been published we have not included it in this paper but have referred to the published figure in the revised text.*

10. l.116: give a range for the meridional gradient
    *The meridional gradient is 50 $Wm^{-2}1000km^{-1}$. This has been added to the text.*

11. Figures 1 and 2: Add Qsw,Qlw, etc. to the captions (as in other figures)
    *Added.*

12. l.143: check the figure number
    *Corrected.*

13. l.152: SST tendencies are discussed in the next section
    *Reference to SST tendency analysis has been removed.*

14. l. 163: 'westward direction': as the composites are rotated it is hard to say where the west is.
    *'Westward direction' has been changed to 'behind the cyclone'*

15. L 159:164: how much are sensible and latent heat fluxes in summer different to those in winter in previous studies (in Tilinina et al. 2018 and Rudeva and Gulev 2011)? From this you may possibly deduce a potential effect of cyclones on SST in winter (which can also be estimated directly in another paper)
    *We have not yet performed this analysis for any other seasons but this is a good suggestion and we will consider this in our future work.*

16. l.165: this sentence suggests that wind should also be shown in fig 5b
    *The text about the winds now only refers to figure 5a.*

17. Fig 3b and d: fix colours in the colour scale (blue - negative, yellow/red - positive)

    *We feel that the colourbar is clear so have not changed the figure.*

18. Fig.4: I'd comment that positive values are into the surface in the caption.

    *This is already mentioned in the caption so we have not made any changes.*

19. Fig.5: add 'air' to panel (a) caption

    *Added.*

20. l.225-232: this paragraph should be in Methods

    *The section describing the cyclone masking has been moved to the methods section.*

21. Fig.9: The relative sizes of the circles are wrong: if the big circle is 30 deg, as the small circle has a 14 deg radius.

    *The schematic has been re-drawn to better reflect the relative size of the circles.*

22. l.237: I do not get why the mask shows the cyclone along the trailing cold front. It suggests that cold fronts always extend along the cyclone centres in the last 30 hours. If that is your assumption, that needs to be proved. As I said at the beginning of the review, I think you need objectively identified cold fronts instead of what has been invented here.

    *We have attempted to show that cold fronts roughly extend along the cyclone centres in the last 30 hours by showing 2 examples of the masking application in figure 10. We decided to use a cyclone tracking algorithm rather than a cold front tracking algorithm because it is the advection of cold dry air behind the cold front which creates the anomalous surface flux. Identifying relatively cold and/or dry airmasses without the co-location with cyclonic winds would not result in large heat flux.*

23. Fig 10: Maybe swap the panels to have 24 Dec on the right and 20 Dec on the left

    *We are unsure why changing the ordering of the figure would improve the clarity of the paper so we have not swapped the panels around.*

24. l.245: 14-18% - what variable does it relate to?

    *Following modifications to the text in response to other comments this sentence has been removed.*

25. l249, 250 : fig 11c and 8f, respectively

    *We have changed the figure references.*

26. Fig11: perhaps I missed it, but was QN due to cyclones calculated for all cyclones in 2013/14, or the strongest? Fig. 11c shows SSTQN due to cyclones or any Qn? I think 11c should be due to cyclones only. Can you explain why strong negative anomalies in the west of the North Atlantic (fig. 11c) are not seen in fig. 11d? I'd say that 11d matches well with 11a, which makes sense, but anomalies in the west North Atlantic in 11d are confusing

    *To clarify, $Q_N$ in figure 11(a) was due to all cyclones in 2013/2014 not just the strongest.*

*Figure 11(c) showed $\Delta SST_{QN}$ due to the total $Q_N$ anomaly combined with the clima-tological MLD. In response to a comment from reviewer 3 we now use the monthly varying MLD for 2013/2014.*

*The SST tendency anomaly for the 2013/2014 season is determined by subtracting the climatological SST tendency from the 2013/2014 SST tendency. The SST tendency anomaly can be separated into the anomaly associated with (i) anomalous $Q_N$ (term 1 in equation 1), (ii) anomalous MLD (term 2 in equation 1) and (iii) anomalous en-trainment through the base of the mixed layer, $Q_{ENT}$ (term 3 in equation 1). We refer to the sum of these quantities as the SST tendency anomaly due to air-sea interactions (ASI), $\Delta SST'_{ASI}$, given by;*

$$\Delta SST'_{ASI} = \frac{Q_N^i - \overline{Q_N}}{\rho c_p \overline{h}} + \frac{Q_N^i}{\rho c_p}\left(\frac{1}{h^i} - \frac{1}{\overline{h}}\right) + \frac{Q_{ENT}^i - \overline{Q_{ENT}}}{\rho c_p \overline{h}}, \tag{1}$$

*where i represents the 2013/2014 values and the overbar represents the 1989-2015 cli-matological values. Since we have no measurements of the entrainment flux anomaly across the ocean boundary layer it is estimated to be 20% of the surface $Q_N$ anomaly (Stull (1988)). Neglecting contributions made by wind driven turbulence.*

*$\Delta SST'$ due to anomalous $Q_N$ is shown in figure 2(a). This closely resembles the $Q_N$ anomaly (figure 11a in the original paper) with anomalous cooling in the mid-North Atlantic where the flux are negative, and anomalous warming (less cooling than clima-tology) in the Gulf Stream and Norwegian Sea regions. $\Delta SST'$ due to anomalous MLD is shown in figure 2(b). This has the opposite pattern to figure 2(a) since larger negative $Q_N$ results in deepening of the MLD via mixing by negatively buoyant water. Thus the large $\Delta SST'$ in the west-North Atlantic are the result of the MLD being anomalously shallow in the 2013/2014 season. The sum, $\Delta SST'_{ASI}$, accounts for 68% of the total $\Delta SST'_{TOT}$ in 2013/2014.*

*We have also calculated both $Q_N$ and $\Delta SST'_{ASI}$ due to the environmental flow and when cyclones are present. Cyclones embedded within the environmental flow are associated with 68% of the total $Q_N$ anomaly, more than double the $Q_N$ due to the environmental flow anomaly only (32%). However, due to significant compensation between $\Delta SST'$ due to anomalous $Q_N$ (figure 3(a)) and $\Delta SST'$ due to anomalous MLD (figure 3(c)) the $\Delta SST'_{ASI}$ when cyclones are present (figure 3(e)) accounts for 41% of the observed $\Delta SST$ compared to 28% when cyclones are not present. Sections 4.1 and 4.2 have been re-written to clarify these points.*

27. l.255: is it entrainment of the cold air?
    *This sentence refers to the entrainment of cold water into the ocean mixed layer from below. This has been clarified in the text.*

28. l.265: As the mask stretches backwards from the cyclone centre, it captures the cold sector. However, the effect of the warm sector remains not assessed (which can also be done if warm fronts are identified).
    *The cyclone masking methodology was designed to capture the anomalous flux occurring behind the cold front and therefore the reviewer is correct, the effect of the warm sector*

[Figure]

Figure 2: Anomalous SST tendency due to 2013/2014 (a) $Q_N$ anomaly, (b) MLD anomaly and (c) $Q_N$, MLD and entrainment anomaly. (d) Total SST tendency anomaly.

*(outside the 14° radius) is not assessed. This can be seen in the examples in figures 10(e) and (f) which capture the anomalously high flux behind the cold front but not the anomalously low flux ahead of the cold front (in the warm sector). However, it is also clear that the negative anomalies behind the cold front are 2-3 times larger in magnitude than the positive anomalies in the warm sector. We have tested the sensitivity of the results to increasing the mask radius to 16° and the contribution of cyclones to the total heat flux anomaly in the mid-north Atlantic increases from 68% to 71%. Therefore, making the mask larger, and thus including more of the warm sector, actually increases the contribution from cyclones. The results of the sensitivity test are already reported in the paper so we have not altered the text.*

29. Fig. 7 suggests that the warm sector will have relatively small effect during the max development, but at other stages of cyclone lifecycles the balance might be different. *Figure 7 shows the SST change due to $Q_N$ only at 3 stages in the cyclone lifecycle (max*

*-24, max and max +24). We have also analysed the SST changes at max -48 and max -36. The effect of the warm sector appears to reduce during these very early stages of cyclone development.*

**Technical comments**

1. The word 'flux' is often used in plural form (e.g., flux occur). My preference is either to say 'flux occurs' or 'fluxes occur'.
   *We have changed 'flux occur' to 'flux occurs' throughout the paper.*

2. l. 73: magnitudes
   *Corrected.*

3. l.101: position is
   *As cyclones is plural, we think that position 'are' rather than 'is' the correct wording.*

4. l.126, 166,173: Figure shows
   *As figure is singular, we think that 'show' rather than 'shows' in the correct wording on these lines.*

5. l.128: 'teh' to 'the'
   *Corrected.*

6. l.135: put comma after 4000Jkg-1K-1
   *Corrected.*

7. l. 137: change 10's to 10s
   *Corrected.*

8. l.147: remove 'are'
   *Removed.*

9. Fig 7: 'Normalised' and 'negative' should start with a small letter
   *Corrected.*

10. l.246: remove 'is'
    *Removed.*

Stull R.B. (1988) Convective Mixed Layer. In: Stull R.B. (eds) An Introduction to Boundary Layer Meteorology. Atmospheric Sciences Library, vol 13. Springer, Dordrecht

[Figure]

Figure 3: 2013/2014 anomalous SST tendency associated with (a) $Q_N$ anomaly due to cyclones, (b) $Q_N$ anomaly due to not associated with cyclones, (c) MLD anomaly when cyclones present, (d) MLD anomaly when cyclones not present, (e) sum of $Q_N$, MLD and entrainment anomalies when cyclones present and (f) sum of $Q_N$, MLD and entrainment anomalies when cyclones not present.

---

## Author Comment (AC1)

H. F. Dacre, S. A. Josey, A. L. M. Grant

December 20, 2019

**Reply to reviewer 1**

We would like to thank the reviewer for their comments on the paper. Below, the reviewers comments are in black and the responses in blue italics. Changes to the paper are shown in red in the revised paper.

**General comment**

The paper documents the sea surface cooling by extratropical cyclones and its impact on the 2013/2014 winter SST in the mid North Atlantic. The conclusions and interpretations are adequately supported for the most part. The paper is well written and conclusions are concise and clear.

*Thank you.*

**Specific comments**

1. Does the warming tendency in the warm sector has any effect on SST? In Section 4.1, the cyclone mask is created so as to encompass the cold front and the cyclone center. Does this method include the warm sector properly?

   *The cyclone masking methodology was designed to capture the anomalous flux occurring behind the cold front and therefore the reviewer is correct, the effect of the warm sector (outside the $14°$ radius) is not assessed. This can be seen in the examples in figures 10(e) and (f) which capture the anomalously high flux behind the cold front but not the anomalously low flux ahead of the cold front (in the warm sector). However, it is also clear that the negative anomalies behind the cold front are 2-3 times larger in magnitude than the positive anomalies in the warm sector. We have tested the sensitivity of the results to increasing the mask radius to $16°$ and the contribution of cyclones to the total heat flux anomaly in the mid-north Atlantic increases from 68% to 71%. Therefore, making the mask larger, and thus including more of the warm sector, actually increases the contribution from cyclones. The results of the sensitivity test are already reported in the paper so we have not altered the text.*

2. The authors focus on the 2013/2014 winter, but I expect that cyclones could play an important role even in other years. The authors might want to estimate cyclones contribution to the winter climatology of the net heat flux using your cyclone masking technique. It would develop a much deeper understanding of the cyclones role.

*This is an excellent suggestion. We have started to apply our cyclone masking technique to other years and seasons. However, including this analysis would increase the length of the paper significantly. Therefore, we will publish this work as a separate publication to avoid a very long paper.*

3. In addition to the strength and number of cyclones, the propagation speed is probably also important for the cooling. The high fraction of time of cyclone mask in 2013/2014 around the UK seems to be partly due to the stagnation of cyclones (Fig. 8).

   *The reviewer is right in their interpretation of the high mask fraction over the UK in the 2013/14 season. Towards the end of the storm track the cyclones slow down becoming quasi-stationary. The effect of propagation speed is taken into account in the masking methodology since multiple timesteps for a single cyclone will contribute to the seasonal climatological cyclone-related $Q_N$. This explanation has been added to section 4 of the revised paper.*

4. Is the anomalously zonal storm track in 2013/2014 associated with the westerly jet?

   *Yes. As shown in Kendon et al. (2015), the 2013/14 season was associated with an anomalously strong and zonally elongated upper-level westerly jet. This extra information has been added to the text in section 4.*

5. The distribution of the Qn anomaly in Figure 8f is different and shifted from that of the cyclones in Figure 8d. Why are they different?

   *As shown in figure 7, the maximum net surface heat flux occurs to the rear of the cyclone centre, typically to the north-west. Therefore, we expect the anomalous net surface heat flux to be to the north-west of the anomalous storm track activity.*

6. The anomalous Qn not associated with cyclones in Figure 11b still has a tripole pattern. So do you think that the tripole pattern has basically nothing to do with cyclones?

   *This is an interesting point. Since the $Q_N$ anomaly pattern when cyclones are not present is similar to that when cyclones are present we conclude that the environmental flow anomaly in 2013/2014 is responsible for generating the tripole of anomalous $Q_N$ values. This pattern is consistent with the anomalous 500hPa geopotential height anomalies over the North Atlantic shown in Bao and Wallace (2015). The role of cyclones embedded within the seasonal flow anomaly is to enhance the negative $Q_N$ anomalies in the mid-Atlantic and reduce the positive $Q_N$ anomalies in the Norwegian Sea. We have re-written the description of this figure in section 4 to make the explanation clearer.*

7. L153-164.rs It is difficult to identify the position of the cold front and warm section in Figure 4. How about plotting the cold and warm fronts? These fronts could be delineated based on Figure 5 or the map of relative vorticity of wind.

   *Cold and warm front position have been added to this figure.*

**Technical comments**

1. L38. of the wind driven currents

   *Changed.*

2. L128. over 6 K over the winter
   *Changed*

3. L135. The density of sea water 1000 kg/m^3 might be acceptable, but the more prac-
   tical value (like 1024 kg/m^3) should be used.
   *In the revised paper $1024 kg/m^3$ has been used for the density of sea water. The conclu-
   sions remain unchanged.*

4. L143. figure 3(a)?
   *Changed.*

5. Figure 4. What do contour lines show?
   *The contour lines show mslp. This has been added to figure 4 caption.*

6. L246. the conclusion does not
   *Changed.*

7. L250. the anomalous Qn
   *Changed.*

8. L250. figure 8(f)
   *Changed.*

Kendon, M. and McCarthy, M., 2015. The UK's wet and stormy winter of 2013/2014. Weather, 70(2), pp.40-47.

Bao, M. and Wallace, J.M., 2015. Cluster analysis of Northern Hemisphere wintertime 500-hPa flow regimes during 1920–2014. Journal of the Atmospheric Sciences, 72(9), pp.3597-3608.

---

## Author Comment (AC3)

H. F. Dacre, S. A. Josey, A. L. M. Grant

December 20, 2019

**Reply to reviewer 3**

I would like to thank the reviewer for their comments on the paper. Below the reviewers comments are in black and the responses in blue italics. Changes to the paper are shown in red in the revised paper.

**General comment**

The Authors present an interesting analysis using ERA-Interim data to address the question how extratropical cyclones influence the SST in the Atlantic. They showcase one particular year that featured a significant SST anomaly and try to attribute a large fraction of this anomaly to anomalous cyclone activity in the same winter. The manuscript is well written and the figures are clear, though the panel labels are sometimes difficult to see as they are on top of shaded figures. Overall, the paper presents a valuable contribution to the field and employs a novel diagnostic to attribute the surface fluxes to individual cyclones.

*Thank you.*

However, there are several points in the paper that need further clarification, which are indicated in the comments below.

**Specific comments**

1. The mixed layer calculation has a caveat, because the authors assume that the depth has no variations throughout the year when they make seasonal budgets. One particular issue with that is that as the mixed layer depth changes, the sea state properties, in particular the stratification below the mixed layer, become important when the mixed layer depth increases. The actual heat content in the mixed layer will depend on the sea state below the mixed layer as well when net surface flux causes mixing. The entrainment of sea water below the column would need to be considered when the fluxes imply a net change in mixed layer depth. It would thus be interesting if the authors also show the seasonal tendency of the mixed layer depth in figure 3, not only the tendency in SST. Given the actual change of mixed layer depth together with the ocean stratification below the mixed layer could yield an estimate of the entrained energy into the changed mixed layer from below. This additional term in the heat budget could be accounted for and contrasted with the net surface forcing of the SST tendency.

*Figure 1 shows the climatological MLD and MLD seasonal tendency. The MLD and*

*MLD tendency patterns are very similar with greatest deepening of the mixed layer occurring where the average MLD is deepest. We have not added the additional figure to the paper but added that 'On average the MLD deepens by 50% between December and February outside the deep convection regions' to the paper text.*

*The reviewer is correct that the entrainment of sea water at the base of the ocean mixed layer is important. Figure 2(a) shows the 2013/2014 SST tendency anomaly, $\Delta SST'$, that is associated with anomalous $Q_N$. As expected $\Delta SST'$ due to anomalous $Q_N$ closely resembles the $Q_N$ anomaly (shown in figure 9(f) in paper) with anomalous cooling in the mid-North Atlantic where the flux are negative, and anomalous warming (less cooling than climatology) in the Gulf Stream and Norwegian sea regions. Small differences are due to spatial inhomogeneity in the North Atlantic climatological MLD. Figure 2(b) shows the 2013/2014 $\Delta SST'$ that is associated with anomalous MLD. The 2013/2014 MLD is shallower than the climatological average over much of the domain, particularly near the Gulf Stream region, and deeper than climatology in the mid-Atlantic region. In the mid-North Atlantic the enhanced negative $Q_N$ results in negative buoyancy and mixing, deepening the MLD. Thus, the surface flux decreases the temperature over a deeper layer of the ocean than usual which reduces the direct SST cooling due to $Q_N$. At the same time, the increased MLD entrains colder water at the base of the ocean mixed layer which cools the surface indirectly. This effect is estimated to be 20% of the $Q_N$ anomaly (Stull, 2012). Neglecting contributinos made by wind driven turbulence. Figure 2(c) shows the sum of the SST tendency anomaly due to anomalous $Q_N$, MLD and entrainment (referred to as the SST tendency anomaly due to air-sea interactions in the paper, $\Delta SST'_{ASI}$). It shows the same tripole pattern as the $\Delta SST'_{TOT}$ (figure 2(d)) which has an average SST cooling amomaly of -1.0K in the mid-North Atlantic region (black box in figures 2(d)). The largest discrepancies occur along the east coast of North America suggesting that ocean dynamics is responsible for transporting warmer water into these regions via the western boundary currents. In the mid-North Atlantic region, the $\Delta SST'_{ASI}$ accounts for 68% of the observed anomalous cooling in the mid-North Atlantic. This figure and explanation has been added to the paper.*

2. Regarding the methodology of cyclone frequency, it is not clear if every cyclone is counted multiple times for the track densities of if some kind of anti-aliasing was employed. This would also influence how storm track activity is defined, as fewer but slower moving storms would yield a higher storm activity in terms of cyclone density compared to the same number of cyclones in a season with hizher phase velocity. It would be great if the authors could further clarify how the cyclone track densities were calculated and how exactly one can thus understand an increased activity of cyclones. It would also be of interest if there were more extreme cyclones that particular year of interest, especially as the authors limit their analysis to the more intense systems.
   *In this paper the effect of propagation speed is taken into account in the masking methodololgy since multiple timesteps for a single cyclone contribute to the seasonal climatological cyclone-related $Q_N$. As a result, the high mask fraction over the UK in the 2013/14 season occurred because there were both a higher than average number of cy-*

[Figure]

Figure 1: North Atlantic DJF 1989-2015 (a) mixed layer depth and (b) mixed layer depth seasonal tendency (m).

*clones and because the cyclones slowed down becoming quasi-stationary over the UK. This clarification of the methodology has been added to the text. The analysis for the 2013/2014 season is not limited to intense cyclones as all cyclones are considered.*

3. A large fraction of the fluxes in the Gulf Stream region are associated with cold air outbreaks, of which a significant fraction is not necessarily associated with cyclones in the storm track region. Could the reduced QN fluxes in 2013/2014 south of the Gulf Stream region as well as in the Nordic Seas be thus actually associated with a reduced number of cold air outbreaks? For the Nordic seas, which also feature a significant anomaly in the presented analysis, Papritz and Spengler (J. Clim., 2017) showed that cold air outbreaks account for the larges fraction of the surface fluxes in this region. Thus, the apparent anomalies are most likely mainly attributable to variations in cold air outbreaks and maybe only indirectly or in a reduced way associated with extratropical cyclones. Papritz and Grams (GRL, 2018) investigated the weather regimes associated with cold air outbreaks in the region of interest in the manuscript at hand. It would be interesting to put their findings and the given role of cold air outbreaks on the surface fluxes in the region in context with the presented findings.

   *It is possible that the reduced $Q_N$ flux in the Gulf Stream region and in the Nordic Seas are associated with cold air outbreaks. Indeed, in the revised version of the paper we attribute these positive heat flux anomalies to the environmental flow pattern which was anomalously zonal, potentially reducing cold air outbreaks. Therefore we have added this explanation to the paper and referenced the papers suggested.*

4. In addition to cold air outbreaks, the role of cold fronts for surface fluxes in the Gulf Stream region has also been discussed recently, e.g., Parfitt and Czaja (2016) and other recent studies by the first author. It would be great if the authors could provide further context of the presented work to these studies.

   *Since we have focussed our analysis on the mid-North Atlantic region and not the Gulf Stream we have not included a detailed discussion of the relationship between this work and that presented by Parfitt and Czaja (2016). However, in future studies we will extend this work to other regions so we thank the reviewer for this reference.*

[Figure]

Figure 2: $\Delta SST'$ due to 2013/2014 (a) $Q_N$ anomaly, (b) MLD anomaly and (c) air-sea interaction anomaly. (d) $DeltaSST'_{TOT}$.

5. The method to define the QN with the cyclone masks is not clear enough. It is difficult to follow what is actually summed up. At each time t for a given cyclone, the position of the cyclone and the preceding 30 hours positions are used, but is this done for every timestep in the cyclone evolution? How would this differ to just taking the swath with circles around all cyclone positions along the entire cyclone track? It would be great if the authors could provide further details about the employed method.

*The reviewer is correct. The mask method is performed for every timestep in the cyclone evolution, which is equivalent to taking a swath with circles around all positions along the cyclone track, but only the track in the preceeding 30 hours, not the entire length of the track. We have made this method clearer in the revised paper.*

**Technical comments**

1. P1 L7: The connection between the "cold wake" and "climatological variability" is not quite clear in this sentence. How is the size of the cold wake associated to climatological

variability?

*Here we specifically refer to climatological variability in the SSTs. We have clarified this in the text.*

2. P1 L21: The argument about the role of cold fronts has also been discussed more recently, e.g., Parfitt and Czaja (2016) and other recent studies by the first author. What is the context of the presented work to these studies?

   *Links to more recent work is made in the introduction (lines 46 onwards). We have added a reference to Parfitt and Czaja (2016) in this section.*

3. P2 L29: After citing the study by Zolina and Gulev (2003), the reader is a bit confused about the thus far identified fluxes associated with extratropical cyclones. If there is a controversy, it would be great if the authors could further highlight these conflicting results and possibly indicate as to why they are conflicting or if they will address these contrasting results.

   *We have expanded the description of the results in this paper which suggest at least partial cancellation of the flux anomalies associated with cyclones.*

4. P2 L28" ...of the wind driven...

   *Corrected.*

5. P2 L44: The authors comment on the role of ocean dynamics in the western Pacific, where oceanic advection probably plays a dominant role. However, the reader is left wondering if not similar arguments would also apply to the western Atlantic, the focus of this study, where strong oceanic currents are present. Are there no studies quantifying the role of oceanic anomalies in the western Atlantic? Good if the authors can also comment on the region of their interest in this context.

   *We have included a reference to Buckley et al. (2015) who also find that in the Gulf Stream region, ocean dynamics are important in setting the upper-ocean heat content anomalies on interannual time scales and that air–sea heat flux damp anomalies created by the ocean.*

6. P2 L51: Another, more direct, connection between cold air outbreaks, cyclones, and the low-level baroclinicity in the western Atlantic is provided by Papritz and Spengler (2015) as well as Vanniere et al. (QJ, 2017).

   *Papritz and Spengler (2015) is cited in the previous sentence so we have not added a further citation here.*

7. P5 L128: "the winter"

   *Corrected.*

8. P7 L140: See general comment about change of mixed layer depth throughout season. Some additional discussion about the influence of mixing and entrainment in the ocean would be valuable.

   *See response to general comment 1.*

9. P7 L144: "heat fluxes occur"

   *Changed to 'heat flux occurs'.*

10. P7 L147-149: This is also the argument of a recent study by Ogawa and Spengler (2019), who also emphasized the role of synoptic eddies on the climatological fluxed in the mid and higher latitudes.
*Thank you for this reference, we were not aware of this paper. A citation to this work has been added.*

11. P9 L183: "the cyclone lifecycle"
*Corrected.*

12. P11 L203: "the surface flux"
*Corrected.*

13. P12 L216: It is not necessarily obvious from the referenced figures that the storm track was more active, see general comment on cyclone track densities.
*See response to specific comment 2.*

14. P13 L223: It is difficult to see how the QN anomaly and the storm track anomaly is "consistent". There appear to be more cyclones detected over the Gulf Stream region in the anomalous winter, though the net negative QN fluxes in this region appear to be reduced when compared to climatology. How can this be reconciled with the previous findings of the cyclone relative QN fluxes and SST changes?
*We agree that the relationship between the $Q_N$ anomaly and storm track anomaly is not clear close to the continental regions where ocean dynamics are dominant. For this reason we have chosen to focus on the mid-Atlantic region only in the paper. We have re-written the text to emphasise that the anomaly in the mid-Atlantic region is consistent with the shift in the storm track, with cyclones travelling more zonally from the US towards western Europe rather than north-eastwards towards Iceland.*

15. P14 L225 and following: The methodology is not quite clear, see also general comments.
*See response to general comment 2.*

16. Fig. 9 caption: "red crosses show"
*Corrected.*

17. P15 L246: "conclusion does not"
*Corrected.*

18. P16 L250: It is not clear that the results indicated in this paragraph consider the data based on the cyclone swaths from the previous section.
*We have re-written this section of the paper to respond to comments from reviewer 2.*

19. P16 L254: The actual percentage of the SST difference cannot be really directly contributed to the fluxes, as it is a mix of local fluxes and advection, as well as entrainment from below that caused the total change. There can be compensating effects that cannot be accounted for in such a crude attribution without actually calculating a full budget considering all tendency terms.
*We agree that several factors contribute to the SST tendency anomaly and that they might be compensating. We have estimated the SST tendency anomaly due to (i)*

*anomalous $Q_N$ , (ii) anomalous MLD and (iii) anomalous entrainment. Therefore, we have performed a more complete analysis of the air-sea interactions and indeed there are compensating effects which are now described in the revised paper.*

20. P17 L262: Can the authors comment further on the relative contributions of potential other effects that make the attribution to individual cyclones difficult?
*We have decomposed the total SST tendency anomaly due to $Q_N$ into three components (see figure 1). We have attributed the difference between the sum of these components (referred to as air-sea interactions) and the total SST tendency anomaly to be due to advection. We note that there are significant assumptions in this method but are confident that the main conclusion that cyclones enhance SST cooling in the mid-North Atlantic region is robust.*

21. P17 L266: The statement about "higher than average cooling" appears to be rather regionally confined and there were also larger areas where this particular season featured reduced air-sea heat exchange. The authors should comment on this complex structure and put it in context to the observed cyclone distribution. Especially the western Atlantic area with reduced fluxes appears difficult to explain given the increased number of cyclones (Fig. 8f, 11d).
*Over almost the entire domain cyclones decrease the SSTs. The reduced air-sea heat exchange over the Gulf Stream and Norwegian Sea is controlled by the environmental flow anomaly. This explanation has been added to the paper text.*

Buckley, M.W., R.M. Ponte, G. Forget, and P. Heimbach, 2015: Determining the Origins of Advective Heat Transport Convergence Variability in the North Atlantic. J. Climate, 28, 3943–3956, https://doi.org/10.1175/JCLI-D-14-00579.1

Stull R.B. (1988) Convective Mixed Layer. In: Stull R.B. (eds) An Introduction to Boundary Layer Meteorology. Atmospheric Sciences Library, vol 13. Springer, Dordrecht

---

## Referee Report (RR1)

Second review of WCD-2019-6:

"Extratropical cyclone induced sea surface temperature anomalies in the 2013/14 winter"
by
Helen F. Dacre, Simon A. Josey, and Alan L. M. Grant

**Recommendation: Accept with minor revisions**

**General Comments:**

I appreciate the authors' detailed response to all my comments and find all my concerns adequately addressed and thereby recommend the manuscript for publication after the very minor comments below have been addressed.

**Minor Comments:**
Page and line numbers refer to the ones on the manuscript.

L58-59: As pointed out previously, the association of low-level baroclinicity and surface fluxes has been demonstrated by Papritz and Spengler (2015). The authors' response states that this paper is cited, though it appears to be still absent in the manuscript.

L146-147: The authors refer to equation (2) before the equation is introduced. Consider rearrangement so that the equation is introduced before discussing the different terms in it.

**Reference:**
Papritz, L., and T. Spengler, 2015: Climatological analysis of the slope of isentropic surfaces and its tendencies over the North Atlantic. Quart. J. Roy. Meteor. Soc., 141, 3226-3238, doi:10.1002/qj.2605